# Newton-PINet: A fast physics-informed neural network with Newton linearization for meta-learning nonlinear PDEs

## Abstract

Scientific machine learning has opened new avenues for solving parameterized partial differential equations (PDEs), enabling models to learn a family of PDEs and generalize to unseen instances. In this context, data-driven operator learning methods typically require large training datasets, while physics-informed neural networks (PINNs) suffer from difficult optimization and limited generalization, especially for nonlinear PDEs. We propose Newton-PINet, a physics-informed network enhanced by Newton linearization, offering an effective meta-learning framework for nonlinear PDEs. Newton-PINet (i) employs a physics-informed multilayer network with skip connections, where the output-layer weights are solved by least squares; (ii) adopts a two-stage learning strategy that first leverages gradient-based training to learn robust representations from the available training tasks, and then performs gradient-free fine-tuning on the output layer for fast task-specific generalization; and (iii) incorporates a Newton linearization method to speed up the least-squares iteration for nonlinear PDE problems. On a challenging nonlinear reaction-diffusion benchmark, Newton-PINet achieves up to three orders of magnitude lower relative error than recent neural solvers, while using 16× fewer training tasks and over an order of magnitude less training time (under 2 minutes versus several hours). This work advances the meta-learning of PINNs toward data-efficient, fast, and generalizable physics solvers. The datasets and code are provided in the supplementary material.

---

[1]Anonymous Institution, Anonymous City, Anonymous Region, Anonymous Country. Correspondence to: Anonymous Author <anon.email@domain.com>.

Preliminary work. Under review by the International Conference on Machine Learning (ICML). Do not distribute.

## 1. Introduction and related works

Solving partial differential equations (PDEs) is fundamental across diverse fields, including fluid dynamics, climate modeling, materials science, and biophysics (Karniadakis et al., 2021). Traditional numerical solvers for PDEs often incur prohibitive computational costs, particularly when repeated evaluations are required in applications such as design optimization, uncertainty quantification, real-time control, and modeling of complex physical systems. In recent years, scientific machine learning has emerged as a powerful paradigm, leveraging advances in deep neural networks to deliver fast and accurate approximations of PDE solutions, thereby enabling high-fidelity simulations and real-time predictive capabilities that were previously unattainable (Cai et al., 2021a; Cuomo et al., 2022). Recent advances in deep learning for PDEs can be broadly categorized into: data-driven operator learning, physics-informed deep learning, and meta-learning physics-informed neural networks (PINNs).

**Data-driven operator learning:** Data-driven operator learning focuses on training neural networks to directly map input functions, such as PDE parameters or initial conditions, to their corresponding PDE solutions. Well-known examples of this approach include DeepONet (Lu et al., 2021), Derivative-enhanced-DeepONet (Qiu et al., 2024), Transformer-DeepONet (Wei et al., 2025b), Fourier neural operators (FNO) (Li et al., 2020), Factorized-FNO (Tran et al., 2023), and Decomposed-FNO (Li & Ye, 2025). Once trained, these models can generalize to new parameter settings (e.g., new initial conditions), enabling fast prediction of complex dynamical behaviors. However, they lack explicit physics constraints and interpretability, and their generalization depends on adequacy of labeled training data, which are often expensive to obtain (Li et al., 2024). For example, Transformer-DeepONet typically requires at least 1,000 different PDE solutions for adequate performance (Wei et al., 2025b).

**Physics-informed deep learning:** An alternative paradigm, physics-informed deep learning, incorporates governing PDEs and boundary conditions into the training loss function (Cai et al., 2021b; Wong et al., 2022; Wei et al., 2025a). By penalizing violation of physical laws, these models can

be trained effectively even in data-sparse regimes, often requiring only initial or boundary conditions. Representative approaches include physics-informed variants of operator learning, such as physics-informed DeepONet (Wang et al., 2021) and physics-informed neural operators (PINO) (Li et al., 2024). Another prominent method is PINN (Raissi et al., 2019), which learn mapping from spatiotemporal coordinates to PDE solution for a given system. While these methods alleviate the need for large labeled datasets, their reliance on physics-informed objectives often results in prohibitively long training times, due to highly nonconvex loss landscapes caused by the stiff PDE constraints (Krishnapriyan et al., 2021; Chiu et al., 2022; Wang et al., 2023), especially for nonlinear PDEs. For example, training physics-informed DeepONet on a family of Burgers' equations can take more than 9 hours (Wang et al., 2021), while PINNs may require over 24 hours to solve a single Kuramoto-Sivashinsky equation (Wang et al., 2025).

**Meta-learning PINNs:** Adapting physics-informed neural networks (PINNs) to new PDEs or parameter settings typically requires costly retraining. Transfer learning strategies accelerate convergence by reusing knowledge from related tasks (Wong et al., 2021; Wang et al., 2022). Parameterized PINNs ($P^2$INNs) further incorporate problem parameters as network inputs, enabling a single model to represent families of PDEs without labeled data while preserving physical consistency (Cho et al., 2024). However, $P^2$INNs still suffer from the optimization challenges of physics-based loss functions, and their adaptation to new tasks remains heavily reliant on gradient-based fine-tuning which is often slow to converge, especially for nonlinear PDEs. Meta-learning methods aim to enable rapid adaptation to unseen PDEs with limited data by learning parameter initializations or update rules (Penwarden et al., 2023; Wong et al., 2025). MAML-style approaches learn transferable parameter priors (Liu et al., 2022), while other methods leverage latent codes or meta-optimizers for task-specific adaptation (Huang et al., 2022; Cho et al., 2023). To avoid gradient-based adaptation, Baldwinian-PINN introduces a gradient-free meta-learning strategy based on neuroevolution, evolving shared representations and fine-tuning the output layer via least squares (Wong et al., 2023). While highly efficient for linear PDEs, its single-layer architecture limits nonlinear expressiveness. Moreover, its Picard (lagging-of-coefficients) iteration exhibits only linear convergence, leading to slow solution refinement for nonlinear problems (Pletcher et al., 2012; Sheu & Lin, 2005; Chiu et al., 2008).

In summary, existing deep learning approaches for solving nonlinear PDEs remain constrained by substantial data requirements and/or slow, optimization-heavy adaptation. For this reason, we propose a novel meta-learning PINN framework, termed **Newton-PINet**. The main contributions are as follows:

- We introduce a physics-informed multilayer network with skip connections from early hidden layers to the output, where the final-layer weights are computed using least-squares approach (Tikhonov regularization). This skip-connected Tikhonov regularization PINN architecture improves the model's nonlinear representation capabilities.

- We employ a two-stage learning strategy, where the first stage uses a gradient-based method to meta-learn the nonlinear hidden layers' network weights and essential hyperparameters for task-specific adaptation in an unsupervised or few-shot manner. At inference (test time), task-specific adaptation is confined to the output layer, which can be rapidly updated in a gradient-free manner using Tikhonov regularization.

- We integrate Newton linearization into the Tikhonov-regularized PINN and demonstrate, through a mathematical derivation, its equivalence to the classical Newton method, thereby preserving its characteristic quadratic convergence. The Newton linearization accelerates nonlinear least-squares convergence, providing a more efficient alternative to the traditional Picard approach during both meta-learning and inference. We therefore refer to the proposed physics-informed network with Newton linearization as Newton-PINet.

Results on diverse nonlinear PDE benchmarks show that, compared with recent neural solvers, Newton-PINet significantly reduces the reliance on training data while achieving fast and accurate generalization to new PDE tasks through physics-consistent fine-tuning.

## 2. Preliminaries

For simplicity, consider a nonlinear PDE with spatial variable $x$, time $t$, and solution $u$ defined on $\Omega \times [0, T]$:

$$\text{PDE:} \quad \mathcal{N}_\theta[u(x,t)] = q(x,t) \quad x \in \Omega, \ t \in [0,T], \quad (1a)$$

$$\text{IC:} \quad u(x, t=0) = u_0(x) \quad x \in \Omega, \quad (1b)$$

$$\text{BC:} \quad \mathcal{B}[u(x,t)] = g(x,t) \quad x \in \partial\Omega, \ t \in [0,T]. \quad (1c)$$

where the general differential operator $\mathcal{N}_\theta$ can include both linear and nonlinear combinations of the temporal and spatial derivatives and PDE parameters $\theta$, $q(x,t)$ is the source term, $u_0(x)$ is the initial condition (IC), and $g(x,t)$ is the boundary condition (BC).

**Single PDE problem:** Standard PINNs aim to solve a single PDE instance (**a single task**) defined by specific PDE parameters, IC, and BC. PINNs are trained by minimizing the discrepancy between Eq. (1) and the model's prediction.

**Towards generalizable PINNs:** There is growing interest in models capable of generalizing across a set of tasks belonging to some underlying task-distribution $p(\mathcal{T})$, e.g., a

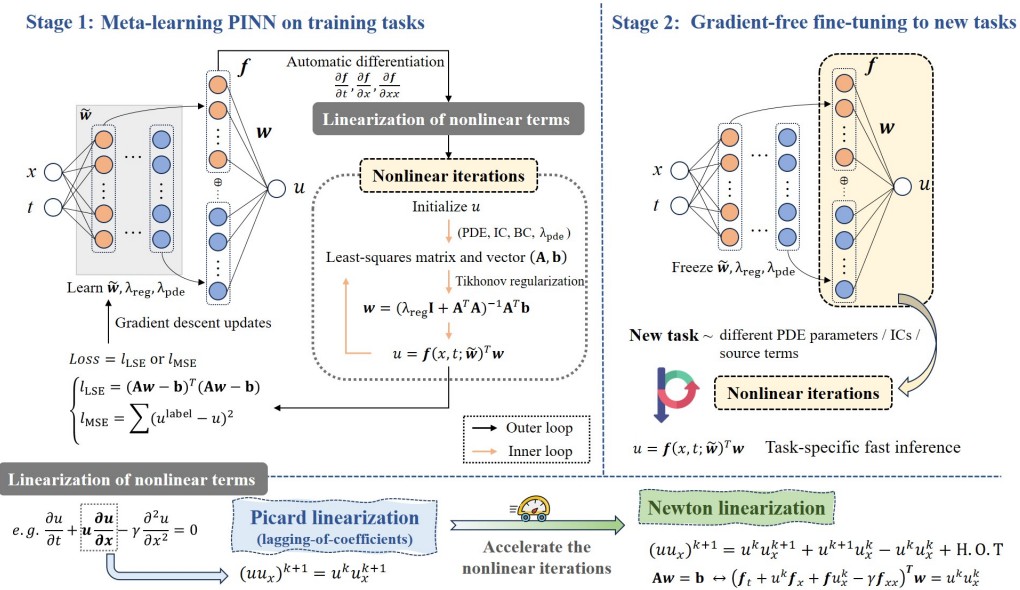

*Figure 1.* Newton-PINet model framework.

family of PDEs spanning different PDE parameters, ICs, and BCs. For meta-learning PINNs, the goal is to learn network initializations using training tasks sampled from $p(\mathcal{T})$ that enable fast, accurate, and physics-aware predictions on unseen scenarios, i.e., any new task from the distribution $\mathcal{T}_i \sim p(\mathcal{T})$ through fine-tuning (Wong et al., 2025).

## 3. Methodology

**Skip-connected Tikhonov regularization PINN:** As opposed to a standard PINN (Raissi et al., 2019), we introduce skip connections from all hidden layers to the output layer, with the output-layer weights computed using Tikhonov regularization (Golub et al., 1999). The skip-connected neural architecture improves expressivity, ensuring stable and accurate least-squares computation by increasing the output-layer width through stacking additional hidden layers, while still maintaining a moderate number of nodes per layer. As shown in Fig. 1, we employ the neural networks with $L + 1$ layers, where the input is $\mathbf{x} = (x, t)$ (layer 0) and the output is $u$ (layer $L$). Each hidden layer $l = 1, \ldots, L - 1$ contains the same number of neurons ($N_n$). Sinusoidal feature embeddings are applied at the first layer (Wong et al., 2022), and all hidden layers use $\sin(\cdot)$ activation functions. The pre-final output $\boldsymbol{f}$ is constructed as the concatenation of all hidden activations (skip connections). All trainable network parameters up to the pre-final layer are denoted $\tilde{\boldsymbol{w}}$. The output $u$ can be computed by $u(\mathbf{x}) = \boldsymbol{f}(\mathbf{x}; \tilde{\boldsymbol{w}})^T \boldsymbol{w}$, where $\boldsymbol{w}$ is the output-layer weights. See Appendix A.1 for a detailed description of the skip-connected model architecture.

The objective of task-specific learning is to determine the best set of $\boldsymbol{w}$ such that $u(\mathbf{x}) = \boldsymbol{f}(\mathbf{x}; \tilde{\boldsymbol{w}})^T \boldsymbol{w}$ satisfies the PDE, IC, and BC, for a task $\mathcal{T}_i \sim p(\mathcal{T})$. This leads to a physics-constrained least-squares problem (Tikhonov regularization): $\arg\min_{\boldsymbol{w}} \|\mathbf{A}\boldsymbol{w} - \mathbf{b}\|_2^2 + \lambda_{\text{reg}} \boldsymbol{w}^T \boldsymbol{w}$. Here $\mathbf{A}\boldsymbol{w}$ is obtained by substituting the model's output into the left-hand side of Eq. (1) at a given set of collocation points, $\mathbf{b}$ denotes the corresponding right-hand side of Eq. (1), and $\lambda_{\text{reg}}$ is the Tikhonov regularization parameter whose proper setting improves the conditioning of the problem. The system has a closed-form solution, either $\boldsymbol{w} = (\lambda_{\text{reg}}\mathbf{I} + \mathbf{A}^T\mathbf{A})^{-1}\mathbf{A}^T\mathbf{b}$ for over-determined system or $\mathbf{A}^T(\lambda_{\text{reg}}\mathbf{I} + \mathbf{A}\mathbf{A}^T)^{-1}\mathbf{b}$ for under-determined one, enabling fast, gradient-free updates. Appendix A.2 provides the physics-based least-squares formulation with implementation details.

Note that Tikhonov regularization serves as a linear solver, which can obtain the solution in a single step for linear PDEs (Wong et al., 2023). For nonlinear PDEs, the nonlinear terms must be linearized so that the system can be cast into a linear form. An initial guess is then required, followed by a few **nonlinear iterations** (a process that can also be interpreted as gradient-free fine-tuning) to update $\boldsymbol{w}$ toward the optimal solution.

Figure 1 illustrates the proposed Newton-PINet model framework, which consists of two stages: (i) meta-learning on training tasks and (ii) gradient-free fine-tuning on new tasks.

**Meta-learning on training tasks:** The meta-learning objective is to optimize the network weights $\tilde{w}$, importance hyperparameter $\lambda_{\text{pde}}$ of PDE loss relative to IC/BC, and the regularization parameter $\lambda_{\text{reg}}$, collectively denoted $\Theta$, to learn task-agnostic representations spanning a family of PDEs and enable fast generalization to unseen tasks requiring only Tikhonov regularization update. In our framework, the outer loop of meta-learning updates the learnable parameters via gradient descent, while task-specific adaptation in the inner loop, or generalization, is performed through gradient-free Tikhonov regularization applied to the output layer weights. The meta-learning objective (outer-loop loss) can be either the physics-based least-squares error $l_{\text{LSE}}(\boldsymbol{w}^*) = \|\mathbf{A}\boldsymbol{w}^* - \mathbf{b}\|_2^2$, or the data-driven mean squared error $l_{\text{MSE}}(\boldsymbol{w}^*) = \frac{1}{n}\sum_s (u_s^{\text{label}} - \boldsymbol{f}(\mathbf{x};\tilde{\boldsymbol{w}})^T \boldsymbol{w}^*)^2$ given labeled data $\{u_s^{\text{label}}\}_{s=1}^n$ where $n$ is the total number of collocation points (PDE residual, IC, and BC). Here, Tikhonov regularization is used to compute the optimal task-specific output-layer weights $\boldsymbol{w}^*$, enabling the model to specialize to any realization of the task. $l_{\text{LSE}}$-based learning is termed **unsupervised learning**. $l_{\text{MSE}}$-based learning is termed **hybrid learning**, since it couples physics-based Tikhonov updates in the inner loop with data-driven minimization in the outer loop. See Algorithm 1 for the pseudo-code and Appendix A.3 for a detailed mathematical description.

**Gradient-free fine-tuning to new tasks:** After meta-learning, $\Theta$ is fixed. For a new task $\mathcal{T}_i \sim p(\mathcal{T})$ with different PDE parameters or IC/BC, Tikhonov regularization is used to compute the task-specific weights $\boldsymbol{w}$, enabling fast gradient-free adaptation to the new PDE instance. Since this update is independent of gradient-based backpropagation optimizers such as stochastic gradient descent (SGD), the resulting adaptation is extremely fast while remaining physics-compliant. See Algorithm 2 in Appendix A.4 for the pseudo-code.

**Newton linearization:** Note that the Tikhonov regularization applies to linear PDEs, where a linear matrix system can be constructed and solved in a single step. For nonlinear PDEs, the nonlinear terms must first be linearized so that the system can be cast into a linear form. The previous approaches typically relied on Picard method (lagging-of-coefficients), which linearizes the nonlinear terms using the solution from the previous iteration and then performs least-squares solves for multiple nonlinear iterations. Although simple, it is only linearly convergent (Dennis Jr & Schnabel, 1996). To address this limitation, we use the Newton linearization approach to approximate the nonlinear term by Taylor expansion around the current iterate. For a function $F$ of one state variable $u$, this gives $F(u^{k+1}) = F(u^k) + \frac{dF}{du}\big|^k (u^{k+1} - u^k) + \text{H.O.T}$, where $k$ is the nonlinear iteration step, and "H.O.T" denotes the higher-order truncated terms of Taylor expansion. For a function of two variables $u$ and $v$, the expansion becomes $F(u^{k+1}, v^{k+1}) = F(u^k, v^k) + \frac{\partial F}{\partial u}\big|^k (u^{k+1} - u^k) + \frac{\partial F}{\partial v}\big|^k (v^{k+1} - v^k) + \text{H.O.T.}$ Based on this principle, the nonlinear terms commonly arising in PDEs can be expressed in a Newton-linearized form amenable to Tikhonov regularization. A detailed mathematical derivation is provided in Appendix A.5, and we demonstrate in Appendix B that the Newton linearization used in the Tikhonov-regularized PINN is essentially equivalent to the classical Newton method, thereby retaining the same quadratic convergence guarantees (Sheu & Lin, 2004; 2005; Chiu et al., 2008). Table 1 summarizes several representative nonlinear terms derived by our work.

*Table 1.* Newton-linearized expressions of several nonlinear terms derived by our work, where $m$ denotes the exponent.

| Nonlinear term | Newton-linearized expression |
|---|---|
| $(u^m)^{k+1}$ | $m(u^k)^{m-1}u^{k+1} + (1-m)(u^k)^m$ |
| $(u^m u_x)^{k+1}$ | $m(u^k)^{m-1}u_x^k u^{k+1} + (u^k)^m u_x^{k+1} - m(u^k)^m u_x^k$ |
| $[\sinh(u)]^{k+1}$ | $\cosh(u^k)u^{k+1} + \sinh(u^k) - \cosh(u^k)u^k$ |
| $[\exp(u)]^{k+1}$ | $\exp(u^k)u^{k+1} + \exp(u^k) - \exp(u^k)u^k$ |
| $[u\ln(u)]^{k+1}$ | $(\ln(u^k)+1)u^{k+1} + u^k\ln(u^k) - (\ln(u^k)+1)u^k$ |

## 4. Experiment results

We compare the performance of the proposed **Newton-PINet** (via Newton linearization), **PINet** (via Picard linearization), vanilla deep neural network (DNN), and recent baseline models (e.g., DeepONet, FNO, PINO), on several representative classes of nonlinear PDEs.

We consider the following representative classes of nonlinear PDEs: (i) Nonlinear convection-type PDEs: $\frac{\partial u}{\partial t} + \beta u^m \frac{\partial u}{\partial x} - \gamma \frac{\partial^2 u}{\partial x^2} + \delta \frac{\partial^3 u}{\partial x^3} + \sigma \frac{\partial^4 u}{\partial x^4} = 0$, where $u(x,t)$ is the state variable, $\beta, \gamma, \delta, \sigma$ are PDE parameters, and $m$ denotes the nonlinearity order. We test three 1D time-dependent problems: Burgers, generalized Korteweg-de Vries (KdV), and Kuramoto-Sivashinsky (K-S) equations. In addition, we consider a 2D lid-driven cavity (LDC) flow governed by the Navier-Stokes equations. These problems span nonlinear convection, high-order dispersion/dissipation, and viscous flow with pressure-velocity coupling, providing canonical benchmarks for assessing generalization across different nonlinearities and dimensions. (ii) Nonlinear forcing-type PDEs: $\frac{\partial u}{\partial t} + \alpha \nabla u - \gamma \Delta u + R(u) = q$, where $R(u)$ denotes a nonlinear reaction operator, which may take polynomial, hyperbolic, exponential, or logarithmic forms, and $q$ represents an external source term. We consider four 1D time-dependent problems: convection-diffusion-reaction (CDR), Klein-Gordon (K-G), hyperbolic heat, and logarithmic heat equations, as well as two 2D problems: Helmholtz and parametric diffusion-reaction equations. These equations represent systems where diffusion and wave propagation in-

teract with nonlinear reaction or source terms, making them central to the modeling of chemical kinetics, heat transfer, quantum fields, and complex reaction-diffusion phenomena in physics and engineering. See Appendix C.1 for detailed problem descriptions, and Appendix C.2 for data generation, error metrics, and computational setup. In Appendix C.3, we provide a summary of meta-learning configurations (Table 4) and model performance (Table 5) across all the PDE problems.

Note that, unless otherwise specified, the meta-learning outer-loop loss function for the Newton-PINet is $l_{\text{MSE}}$ (hybrid learning). We also adopt a temporal domain decomposition strategy within our meta-learning framework to leverage temporal causality as a means of improving accuracy for complex time-dependent PDEs (Wang et al., 2022). Each time block is trained to model only short-term dynamics; however, iterative composition across blocks during inference ensures seamless integration over the full temporal horizon.

### 4.1. Learning to solve PDEs with nonlinear convection term

**Burgers' problems:** $\frac{\partial u}{\partial t} + u \frac{\partial u}{\partial x} - \gamma \frac{\partial^2 u}{\partial x^2} = 0$. We first fix the initial condition as $u(x, 0) = -\sin(\pi x)$, and vary the viscosity parameter $\gamma$ in the range $[0.001, 0.05]$ with an increment of $0.001$ to generate 50 tasks. Among them, 16 are randomly selected for training, and the remaining 34 are used as test tasks. Under each condition, we keep the number of nonlinear iterations (N) for inner-loop fine-tuning in meta-learning stage consistent with those in generalization stage. We then compare Newton-PINet and PINet across different conditions, with the number of nonlinear iterations (N) ranging from 2 to 12. As shown in Fig. 2 (a) and (b), with an increase of N, PINet shows improved convergence during training, but test task errors remain consistently high. In contrast, Newton-PINet achieves significantly faster convergence during meta-learning and achieves low errors on test tasks (lowest error at N = 6). We also record the meta-learning and inference time of both models. As shown in Fig. 2 (c), although Newton-PINet involves additional computations due to Newton linearization, its meta-learning and inference time remain almost unaffected. These results demonstrate that Newton linearization significantly improves the convergence of meta-learning, which also enhances generalization accuracy on test tasks.

In addition to the $l_{\text{MSE}}$ meta-learning loss, we evaluate alternative losses: the least-squares error ($l_{\text{LSE}}$) and the combined loss ($l_{\text{LSE+MSE}}$). As shown in Fig. 2 (e), Newton-PINet achieves the highest test accuracy when trained with $l_{\text{MSE}}$. Since the Tikhonov regularization (inner loop) already enforces the PDE constraints, introducing an additional LSE term in the outer-loop loss can lead to conflict between the two objectives. Our experience shows that the MSE-only meta-objective often provides better generalization in Newton-PINet. In addition, the regularization parameter ($\lambda_{\text{reg}}$) is not manually tuned but meta-learned in this study. Appendix D.1 provides ablation results demonstrating that the meta-learned value is robust across diverse test tasks.

**Model comparison under varying initial conditions:** We evaluate the model on four additional settings with $\gamma \in \{0.1, 0.01, 0.001, 0.0001\}$, where, for each $\gamma$, the initial conditions are sampled from a Gaussian random field $\mathcal{N}(0, 625(-\Delta + 25I)^{-4})$ to generate 50 training and 1000 test tasks. Experiments are conducted independently for each $\gamma$. We compare (i) **unsupervised Newton-PINet** (1 time block) and (ii) **hybrid Newton-PINet** (4 time blocks) against popular neural operator baselines in terms of training time and test error. The **unsupervised** operator considered is Physics-Only DeepONet (PO-DeepONet) (Wang et al., 2021). The **supervised** operators include DeepONet (Lu et al., 2021), Transformer-DeepONet with Trunk net enhanced by Fourier coefficients (T-DeepONet-TF) (Wei et al., 2025b), and FNO (Li et al., 2020), while the **hybrid** operator is PINO (a physics-informed FNO) (Li et al., 2024). Note that PINO uses a training loss that combines both data-driven and PDE-based physics losses. The results are summarized in Table 2. Compared to PO-DeepONet, unsupervised Newton-PINet reduces training time from 9.25h to 291s while improving test accuracy across all $\gamma$ experiments (in a data-absent scenario). Compared to supervised and hybrid neural operators, including the state-of-the-art T-DeepONet-TF and PINO, Newton-PINet consistently achieves superior test accuracy under $\gamma = 0.1, 0.01, 0.0001$, with significantly reduced training cost (10x less time). Our model consistently achieves higher accuracy while requiring the least training time, whether fully unsupervised (without data) or with a few labeled samples.

**Generalized KdV, K-S, and LDC problems:** We further evaluate Newton-PINet on more complex nonlinear PDEs, including the generalized KdV equations with higher-order nonlinear terms $u \frac{\partial u}{\partial x}$, $u^2 \frac{\partial u}{\partial x}$, $u^3 \frac{\partial u}{\partial x}$, the K-S equations with nonlinear term $u \frac{\partial u}{\partial x}$, and the LDC problem governed by the Navier-Stokes equations. These PDEs pose highly challenging benchmarks for PINNs. The state-of-the-art PirateNets+SOAP (Wang et al., 2025) requires over 24 hours of training to solve a single K-S problem and must be retrained for each new problem. In contrast, our **Newton-PINet** completes the meta-learning stage in *under half an hour*, while single-task adaptation takes *less than one second*, demonstrating a clear advantage in computational efficiency. The results (summarized in Appendix C.3 Tables 4 and 5) show that Newton-PINet achieves test errors of MSE $= 1.94 \times 10^{-3}$ on the generalized KdV, MSE $= 3.45 \times 10^{-2}$ on the K-S, and MSE $= 1.67 \times 10^{-4}$

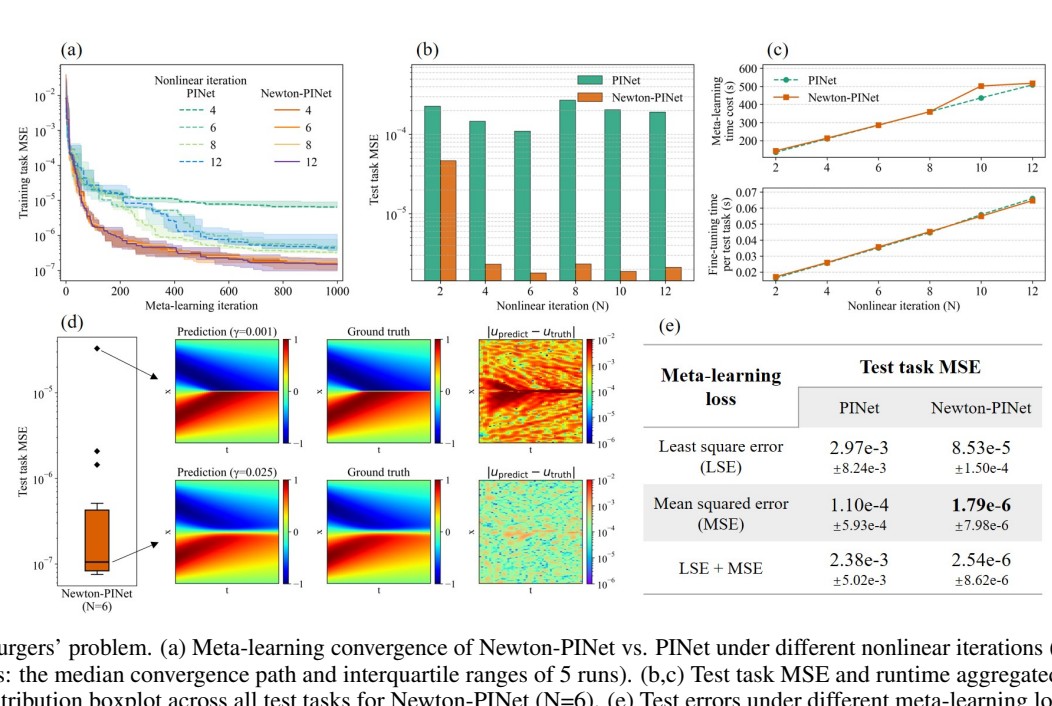

*Figure 2.* Burgers' problem. (a) Meta-learning convergence of Newton-PINet vs. PINet under different nonlinear iterations (N) (lines and shaded areas: the median convergence path and interquartile ranges of 5 runs). (b,c) Test task MSE and runtime aggregated from 5 runs. (d) Error distribution boxplot across all test tasks for Newton-PINet (N=6). (e) Test errors under different meta-learning losses.

on the LDC problem, outperforming PINet and DNN under the same configuration. Figures 5, 6, 7 show the prediction results of these problems. This demonstrates that our Newton linearization can improve the model performance across a variety of nonlinear convective PDEs. However, the initial guess ($u_{\text{guess}}$) in the nonlinear iterations can affect convergence. We conduct empirical ablations in Appendix D.2 to unveil a few strategies to mitigate the sensitivity to initialization. Setting $u_{\text{guess}}$ at all time steps equal to the initial field ($u_{t=0}$) can help provide high accuracy for time-dependent problems, while a temporal block decomposition strategy may be employed for increasingly more complex nonlinear time-dependent systems.

### 4.2. Learning to solve PDEs with nonlinear forcing term

**Convection-diffusion-reaction (CDR) problem:** $\frac{\partial u}{\partial t} + \alpha \frac{\partial u}{\partial x} - \gamma \frac{\partial^2 u}{\partial x^2} - \rho_1 u + \rho_2 u^2 + \rho_3 u^3 = 0$. We generate 40 training and 160 test tasks by varying PDE parameters and initial conditions: $\alpha = 1$, $\gamma \in \{0.005, 0.01, 0.05\}$, $\rho_1, \rho_2 \in \{0, 1, 3, 5\}$, $\rho_3 = 5$, and $u(x, t = 0) = \sum_{j=1}^{J} A_j \sin(l_j x + \phi_j)$, where $J = 5$, $A_j \in [0.1, 0.5]$, $l_j \in \{1, 2, 3, 4\}$, and $\phi_j \in [-\pi, \pi]$. We adopt a 4-time-block decomposition during meta-training and inference to improve accuracy. Figure 3 (a) and (b) show the meta-learning process and test results. It can be seen that PINet converges well during meta-learning, and its test error decreases as nonlinear iteration (N) increases. In contrast, Newton-PINet converges much faster and achieves lower

test error. As shown in Fig. 3 (c) and (d), for a comparable error level, Newton-PINet (N = 4) requires less than half the meta-training and fine-tuning time of PINet (N = 12). These results indicate that Newton linearization can substantially accelerate nonlinear iterations without significantly introducing additional time cost in either meta-learning or inference. More prediction results under various test conditions are provided in Appendix Fig. 8. Note that different time-blocking strategies can affect model inference performance, as discussed in Appendix D.3. As shown in Fig. 3, the predictions exhibit drift at temporal block boundaries. We can mitigate this drift by increasing the initial condition weight, as detailed in Appendix D.4.

**Additional nonlinear forcing-type PDEs:** We evaluate Newton-PINet on several other challenging nonlinear forcing-type PDEs: Klein-Gordon, hyperbolic/logarithmic heat, Helmholtz, and parametric diffusion-reaction equations. The results show that Newton-PINet outperforms the DNN and PINet baselines on nearly all problems (except for the Klein-Gordon problem). Notably, for problems with hyperbolic or logarithmic nonlinearities, the Picard approach used by PINet often fails to converge or leads to unstable training, whereas Newton-PINet remains stable and accurate. Additional results for these problems are provided in Appendix C.3, Table 5, and Fig. 9. We further demonstrate the model's applicability to a wide range of boundary condition types in Appendix D.5.

*Table 2.* Model comparison on the 1D Burgers' problem. The initial conditions for generating all the training and test data are drawn from GRF $\sim \mathcal{N}(0, 625(-\Delta + 25I)^{-4})$. Our Newton-PINet results are computed on a Tesla V100 GPU. The lowest errors are highlighted in bold. "–" denotes results not reported in the references.

| | Model | No. training tasks | No. test tasks | Test relative $L^2$ error (training time) | | | |
|---|---|---|---|---|---|---|---|
| | | | | $\gamma = 0.1$ | $\gamma = 0.01$ | $\gamma = 0.001$ | $\gamma = 0.0001$ |
| Unsupervised | PO-DeepONet (Wang et al., 2021) | 1000 | 1000 | – | 1.38e-2 (9.25h) | 2.16e-1 – | 2.48e-1 – |
| | **Newton-PINet** (Ours) | 50 | 1000 | 1.33e-3 (291s) | 3.51e-3 (291s) | 1.37e-1 (291s) | 2.01e-1 (291s) |
| Supervised | DeepONet (Wei et al., 2025b) | 1000 | 500 | – | 1.17e-2 (2800s) | 2.30e-1 (2620s) | 2.88e-1 (2660s) |
| | T-DeepONet-TF (Wei et al., 2025b) | 1000 | 500 | – | 2.08e-3 (3041s) | **9.81e-3** (2569s) | 1.18e-1 (3466s) |
| | FNO (Li et al., 2020) | 1000 | 200 | 1.39e-2 | – | – | – |
| Hybrid | PINO (Li et al., 2024) | 1000 | 200 | – | 3.80e-3 (1200s) | – | – |
| | **Newton-PINet** (Ours) | 50 | 1000 | **1.13e-3** (133s) | **5.46e-4** (133s) | 4.66e-2 (133s) | **9.15e-2** (133s) |

*Table 3.* Model comparison on the 1D nonlinear reaction-diffusion problem. Baseline results are from (Boudec et al., 2024) (computed on NVIDIA RTX A6000 GPU), while our Newton-PINet results are computed on a Tesla V100 GPU. Time is reported in hours (h), minutes (m), and seconds (s). Best and second-best are bold and underlined, respectively.

| | Model | No. training / test tasks | Test relative MSE | Training time | Inference time |
|---|---|---|---|---|---|
| Unsupervised | PINNs+L-BFGS | | 6.13e-1 | – | 369s |
| | PINNs-multi-opt | | 7.57e-1 | – | 16.5s |
| | PPINNs | 800 / 200 | 3.94e-1 | 4h15m | 0.291s |
| | P$^2$INNs | | 5.69e-1 | 11h | 0.676s |
| | PO-DeepONet | | 4.10e-1 | 3h30m | 0.438s |
| Hybrid | PI-DeepONet | | 7.90e-2 | 3h30m | 0.443s |
| | PINO | 800 / 200 | 4.21e-4 | 1h10m | 0.519s |
| | PI-neural-solver | | 2.91e-4 | 4h30m | 0.284s |
| | **Newton-PINet** (Ours) | 50 / 200 | **1.71e-7** | **119s** | **0.084s** |

**Model comparison on the nonlinear reaction-diffusion benchmark problem:** We further evaluate our method on a widely studied 1D nonlinear reaction-diffusion problem $\frac{\partial u}{\partial t} - \gamma \frac{\partial^2 u}{\partial x^2} - \rho u(1 - u) = 0$ (Krishnapriyan et al., 2021). The benchmark configuration and baseline model results are all taken from (Boudec et al., 2024). The benchmark uses a fixed Gaussian initial condition, with PDE parameters sampled over $\gamma \in [1, 5]$ and $\rho \in [-5, 5]$. The baseline models are grouped into two categories: (i) **unsupervised**, including instance-wise PINNs trained with L-BFGS (PINNs+L-BFGS) or Adam+L-BFGS (PINNs-multi-opt), parametric PINNs (PPINNs) (Boudec et al., 2024) and P$^2$INNs (Cho et al., 2024) that incorporate PDE parameters as inputs, and Physics-Only DeepONet (PO-DeepONet) (Wang et al., 2021); and (ii) **hybrid**, which combine supervised and physics-informed approaches, such as PI-DeepONet (Goswami et al., 2023), PINO (Li et al., 2024), and the physics-informed neural solver (PI-neural-

solver) (Boudec et al., 2024). These baselines were trained with 800 tasks and tested on 200 unseen tasks. In contrast, our hybrid **Newton-PINet** *achieves state-of-the-art performance using only 50 training tasks*, i.e., *16× fewer tasks*, and is tested on 200 unseen tasks using a 4-time-block decomposition. As shown in Table 3, Newton-PINet reaches a test MSE roughly three orders of magnitude lower than the best-performing baseline (PI-neural-solver) while requiring only 119 seconds of total meta-training wall-clock time, compared with several hours for the baselines. These results highlight the *high data efficiency and computational efficiency* of Newton-PINet. Model prediction results are provided in Appendix Fig. 10.

**Comparison with other meta-learning PINN methods:** We further compare the performance of Newton-PINet with state-of-the-art gradient-based meta-learning PINNs (Penwarden et al., 2023) and gradient-free Baldwinian-

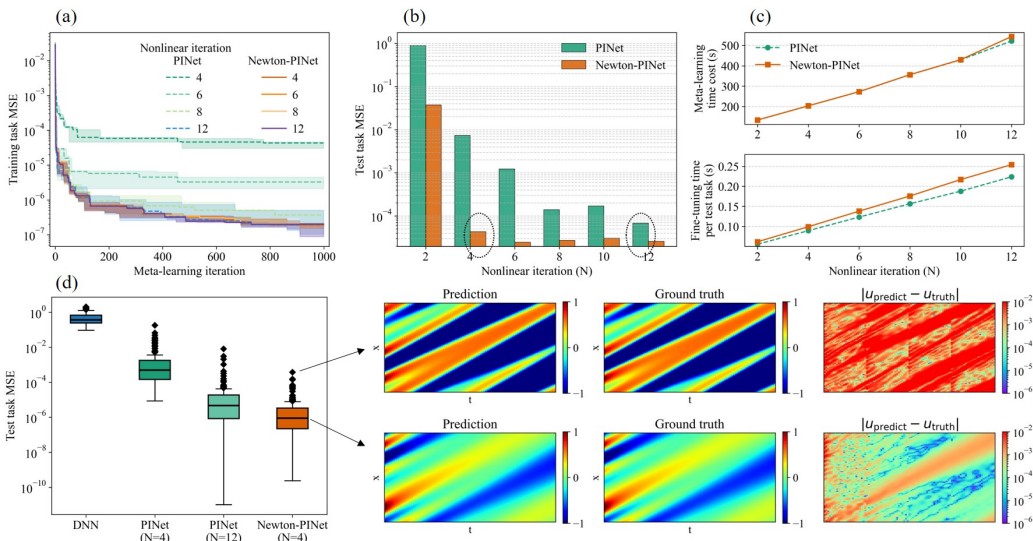

*Figure 3.* Convection-diffusion-reaction (CDR) problem. (a) Meta-learning convergence of Newton-PINet vs. PINet under different nonlinear iterations (N) (lines and shaded areas: the median convergence path and interquartile ranges of 5 runs). (b,c) Test task MSE and runtime aggregated from 5 runs. In (b), the two circled regions correspond to Newton-PINet (N=4) and PINet (N=12), respectively. (d) Error distribution boxplot across all test tasks for different models; the prediction fields correspond to Newton-PINet (N=4).

PINN (Wong et al., 2023). Detailed results are provided in Appendix C.4 and Table 6. On a 2D parametric diffusion-reaction problem, Newton-PINet achieves approximately a *179× improvement in generalization accuracy* while requiring *7000× less fine-tuning time* compared with gradient-based meta-learning PINNs. Moreover, relative to Baldwinian-PINN, Newton-PINet reduces test MSE by 2∼5 orders of magnitude across benchmark problems and shortens the task-specific fine-tuning time by roughly one order of magnitude. These results highlight Newton-PINet's superior generalization performance and computational speed.

In Appendix D.6, we demonstrate that Newton-PINet benefits significantly from skip connections, which enhance robustness to depth, width, and mesh resolution, thereby reducing the need for extensive hyperparameter tuning. In Appendix E, we discuss Newton-PINet's scalability and potential future directions. Briefly, since the features are generated by a neural representation, each row of **A** is dense rather than sparse in the numerical sense. The computational cost primarily depends on the expressiveness of pre-final layer features. By meta-learning a compact yet informative representation, the least-squares solve remains efficient even for large collocation sets.

## 5. Conclusion

Newton-PINet demonstrates robust performance in meta-learning nonlinear PDEs. Our model's strengths can be summarized as follows. **(i) Computational efficiency**:

task adaptation in both meta-learning and inference is performed via least-squares updates of only the output layer. This gradient-free fine-tuning is faster than typical methods such as SGD, requiring just one step for linear PDEs and a few iterations for nonlinear PDEs. The quadratic convergence of Newton linearization further accelerates nonlinear solves. **(ii) Data efficiency**: because the least-squares fine-tuning for new-task generalization is physics-informed, the meta-learning stage of our model requires very few labeled training samples and can converge in only a few epochs. **(iii) Accuracy**: Tikhonov regularization stabilizes the least-squares solve, reducing ill-conditioning and improving generalization accuracy. Altogether, Newton-PINet achieves high generalization accuracy while requiring an order of magnitude fewer training tasks than state-of-the-art baselines. Task-specific inference on new tasks is also orders of magnitude faster compared to gradient-based meta-learning PINNs. These advantages make Newton-PINet a practical and scalable framework for learning large families of nonlinear PDEs in few-shot and real-time scenarios.

## Impact Statements

This paper presents work whose goal is to advance the field of machine learning. There are many potential societal consequences of our work, none of which we feel must be specifically highlighted here.

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

# Appendix

In this Appendix, we provide comprehensive materials to facilitate a deeper understanding of our study.

- Appendix A presents a detailed description of the Newton-PINet model, including the meta-learning and gradient-free fine-tuning framework, and the Newton linearization method.

- Appendix B mathematically shows the quadratic convergence of the Newton linearization.

- Appendix C describes the data generation, computational configurations, and additional experimental results.

- Appendix D reports some ablation studies.

- Appendix E discusses the practical scaling of Newton-PINet.

- Appendix F states the use of large language models (LLMs) in this study.

## A. Newton-PINet model

This study presents a skip-connected Tikhonov regularization PINN model for meta-learning nonlinear PDEs, enhanced with Newton linearization, to enable efficient few-shot learning and fast generalization, hereafter referred to as Newton-PINet.

### A.1. Skip-connected PINN architecture

Building upon the conventional feed-forward MLP architecture of PINNs, we introduce skip connections from all hidden layers to the output layer, with the output-layer weights computed using Tikhonov regularization.

The skip-connected neural architecture enables us to provide better expressivity to ensure stable and accurate least-squares computation by increasing the width of the output layer through the stacking of additional hidden layers, even while maintaining a moderate number of nodes per layer. In addition, the concatenation mechanism can be interpreted as expanding a richer basis space: each added hidden layer introduces new nonlinear features, analogous to incorporating higher-order terms in a polynomial or Chebyshev-type basis, leading to a more expressive representation with reduced truncation effects. A complementary viewpoint is that skip connections also help stabilize optimization—much like in ResNets—by mitigating vanishing-gradient issues in deeper architectures.

This architecture resembles extreme learning machines (ELMs) in its stacked structure (Dong & Li, 2021). However, in traditional ELMs, hidden-layer weights are randomly initialized, which makes training deeper architectures increasingly difficult and can compromise the accuracy of the solution. In contrast, our framework employs a meta-learning strategy to update the hidden-layer weights instead of relying on random initialization. As a result, the network depth can be increased appropriately to improve nonlinear representation capacity while maintaining stable and accurate generalization.

We employ the Tikhonov-regularized PINN with $L + 1$ layers, where the input is $\mathbf{x} \in \mathbb{R}^{D_{in}}$ (layer 0) and the output is $\mathbf{u} \in \mathbb{R}^{D_o}$ (layer $L$). Here, $D_{in}$ and $D_o$ denote the dimensionality of the input and output variables, respectively. Each hidden layer $l = 1, \ldots, L - 1$ contains the same number of neurons ($N_n$). Sinusoidal feature embeddings are applied at the first layer (Wong et al., 2022), and all hidden layers use $\sin(\cdot)$ activation functions. Skip connections from all hidden layers to the output layer are employed to allow each hidden layer to contribute nonlinearly to the final representation without additional weights in the concatenation.

The transformation and activation in the first hidden layer are defined as

$$z_j^1 = \sum_{d=1}^{D_{in}} W_{j,d}^1 x_d + b_j^1, \quad f_j^1(\mathbf{x}) = \sin(2\pi z_j^1), \quad j = 1, \ldots, N_n, \tag{2}$$

where $j$ and $d$ denote neurons in the current and previous layers respectively, $W^1 \in \mathbb{R}^{N_n \times D_{in}}$, and $b^1 \in \mathbb{R}^{N_n}$. We incorporate Sinusoidal feature embeddings in the first layer to enhance the representation of high-frequency components.

For subsequent hidden layers $l = 2, \ldots, L - 1$, we have

$$z_j^l = \sum_{d=1}^{N_n} W_{j,d}^l f_d^{l-1} + b_j^l, \quad f_j^l(\mathbf{x}) = \sin(z_j^l), \quad j = 1, \ldots, N_n, \tag{3}$$

where $W^l \in \mathbb{R}^{N_n \times N_n}$ and $b^l \in \mathbb{R}^{N_n}$.

All trainable network parameters up to the pre-final layer are denoted as $\tilde{\boldsymbol{w}} = [W^l, b^l], \; l = 1, \ldots, L-1$.

The pre-final output is constructed as the concatenation of all hidden activations (skip connections):

$$\boldsymbol{f}(\mathbf{x}; \tilde{\boldsymbol{w}})^T = \left[ f_1^1, \ldots, f_{N_n}^1, \; f_1^2, \ldots, f_{N_n}^2, \; \ldots, \; f_1^{L-1}, \ldots, f_{N_n}^{L-1} \right], \tag{4}$$

which is then connected to the output layer with $D_o$ dimension, so that $\boldsymbol{f}(\mathbf{x}; \tilde{\boldsymbol{w}})^T \in \mathbb{R}^{D_o \times N_n(L-1)}$. Then, we need to flatten it, resulting in $\boldsymbol{f}(\mathbf{x}; \tilde{\boldsymbol{w}})^T \in \mathbb{R}^{1 \times D_o N_n(L-1)}$.

The output $u$ can be computed by

$$u(\mathbf{x}) = \sum_j w_j f_j = \boldsymbol{f}(\mathbf{x}; \tilde{\boldsymbol{w}})^T \boldsymbol{w}, \tag{5}$$

where $\boldsymbol{w}^T = [\ldots w_j \ldots] \in \mathbb{R}^{1 \times D_o N_n(L-1)}$ denotes the output-layer weights.

## A.2. Physics-based least-squares formulation

The objective of task-specific learning is to determine the best set of $\boldsymbol{w}$ such that $u(\mathbf{x}) = \boldsymbol{f}(\mathbf{x}; \tilde{\boldsymbol{w}})^T \boldsymbol{w}$ satisfies the PDE, IC, and BC, for a task ($\mathcal{T}_i$):

$$\begin{align}
\text{PDE:} \quad & \mathcal{N}_\theta[u(x,t)] = q(x,t) \quad x \in \Omega, \; t \in [0, T], \tag{6a} \\
\text{IC:} \quad & u(x, t=0) = u_0(x) \quad x \in \Omega, \tag{6b} \\
\text{BC:} \quad & \mathcal{B}[u(x,t)] = g(x,t) \quad x \in \partial\Omega, \; t \in [0, T], \tag{6c}
\end{align}$$

where $\mathcal{N}_\theta, q, u_0, g$ denote the differential operator with PDE parameters $\theta$, the source term, IC, and BC, respectively.

This leads to a physics-based least-squares formulation. Let $\mathbf{x} = (x, t)$. Given collocation points for PDE residual: $(x_n^{\text{pde}}, t_n^{\text{pde}})$, $n = 1, \ldots, n_{\text{pde}}$; points at the initial time: $(x_n^{\text{ic}}, 0)$, $n = 1, \ldots, n_{\text{ic}}$; and points on the boundary: $(x_n^{\text{bc}}, t_n^{\text{bc}})$, $n = 1, \ldots, n_{\text{bc}}$, together with the loss importance hyperparameter $\lambda_{\text{pde}}$, the following system is obtained:

$$\begin{bmatrix} \cdots & \lambda_{\text{pde}} \cdot \mathcal{N}_\theta[f_j(x_1^{\text{pde}}, t_1^{\text{pde}}; \tilde{\boldsymbol{w}})] & \cdots \\ & \vdots & \\ \cdots & \lambda_{\text{pde}} \cdot \mathcal{N}_\theta[f_j(x_{n_{\text{pde}}}^{\text{pde}}, t_{n_{\text{pde}}}^{\text{pde}}; \tilde{\boldsymbol{w}})] & \cdots \\ \cdots & f_j^i(x_1^{\text{ic}}, 0; \tilde{\boldsymbol{w}}) & \cdots \\ & \vdots & \\ \cdots & f_j(x_{n_{\text{ic}}}^{\text{ic}}, 0; \tilde{\boldsymbol{w}}) & \cdots \\ \cdots & \mathcal{B}[f_j(x_1^{\text{bc}}, t_1^{\text{bc}}; \tilde{\boldsymbol{w}})] & \cdots \\ & \vdots & \\ \cdots & \mathcal{B}[f_j(x_{n_{\text{bc}}}^{\text{bc}}, t_{n_{\text{bc}}}^{\text{bc}}; \tilde{\boldsymbol{w}})] & \cdots \end{bmatrix} \begin{bmatrix} \vdots \\ w_j \\ \vdots \end{bmatrix} = \begin{bmatrix} q(x_1^{\text{pde}}, t_1^{\text{pde}}) \\ \vdots \\ q(x_{n_{\text{pde}}}^{\text{pde}}, t_{n_{\text{pde}}}^{\text{pde}}) \\ u_0(x_1^{\text{ic}}) \\ \vdots \\ u_0(x_{n_{\text{ic}}}^{\text{ic}}) \\ g(x_1^{\text{bc}}, t_1^{\text{bc}}) \\ \vdots \\ g(x_{n_{\text{bc}}}^{\text{bc}}, t_{n_{\text{bc}}}^{\text{bc}}) \end{bmatrix}, \tag{7}$$

which can be compactly written as

$$\mathbf{A}\boldsymbol{w} = \mathbf{b}.$$

Here $\mathbf{A}\boldsymbol{w}$ represents the left-hand side of Eq. (6) at a given set of collocation points, $\mathbf{b}$ denotes the corresponding right-hand side of Eq. (6).

Therefore, using the PDE, IC, and BC, the physics-informed matrix $\mathbf{A}$ and vector $\mathbf{b}$ can be assembled. Concretely, the pre-final representation $\boldsymbol{f}$ is obtained via forward propagation, and its derivatives with respect to $(x, t)$ in $\mathcal{N}_\theta$ are computed through automatic differentiation (AD).

For a single task, i.e., a single PDE instance, the number of rows in $\mathbf{A}$ and $\mathbf{b}$ equals the total number of collocation points, including those for the PDE residual, IC, and BC. The number of columns in $\mathbf{A}$ corresponds to the product of the pre-final feature dimension $N_n(L-1)$ and the output dimension $D_o$. This results in $\mathbf{A} \in \mathbb{R}^{(n_{\text{pde}} + n_{\text{ic}} + n_{\text{bc}}) \times D_o N_n(L-1)}$ and $\mathbf{b} \in \mathbb{R}^{(n_{\text{pde}} + n_{\text{ic}} + n_{\text{bc}}) \times 1}$.

The final output weights $\boldsymbol{w}$ are obtained via Tikhonov regularization:

$$\arg\min_{\boldsymbol{w}} \ \|\mathbf{A}\boldsymbol{w} - \mathbf{b}\|_2^2 + \lambda_{\mathrm{reg}}\,\boldsymbol{w}^T\boldsymbol{w}. \tag{8}$$

where $\lambda_{\mathrm{reg}} \geq 0$ denotes the regularization parameter. Compared with ordinary least squares (OLS), which minimizes only the residual norm $\|\mathbf{A}\boldsymbol{w} - \mathbf{b}\|_2^2$, Tikhonov regularization introduces an additional penalty term $\lambda_{\mathrm{reg}}\,\boldsymbol{w}^T\boldsymbol{w}$. This term suppresses large weights, stabilizes the solution, and improves the conditioning of the problem.

Note that for linear PDE, the closed-form Tikhonov-regularized solution of Eq. (7) can be obtained in a single step:

$$\boldsymbol{w} = \begin{cases} (\lambda_{\mathrm{reg}}\mathbf{I} + \mathbf{A}^T\mathbf{A})^{-1}\mathbf{A}^T\mathbf{b}, & \text{if Eq. (7) is over-determined,} \\[2mm] \mathbf{A}^T(\lambda_{\mathrm{reg}}\mathbf{I} + \mathbf{A}\mathbf{A}^T)^{-1}\mathbf{b}, & \text{if Eq. (7) is under-determined.} \end{cases} \tag{9}$$

For nonlinear PDEs, the nonlinear terms must first be linearized so that the system can be cast in the form of Eq. (7). For example, in the Burgers equation: $u_t + uu_x - \gamma u_{xx} = 0$, we can compute

$$u = \boldsymbol{f}^T\boldsymbol{w}, \quad u_t = \frac{\partial u}{\partial t} = \boldsymbol{f}_t^T\boldsymbol{w}, \quad u_x = \frac{\partial u}{\partial x} = \boldsymbol{f}_x^T\boldsymbol{w}, \quad u_{xx} = \frac{\partial^2 u}{\partial x^2} = \boldsymbol{f}_{xx}^T\boldsymbol{w},$$

where the pre-final representation $\boldsymbol{f}$ and its derivatives can be obtained via forward propagation and AD. However, the nonlinear term $uu_x$ cannot be directly expressed as a combination of $\boldsymbol{f}$ (or $\boldsymbol{f}_x$) and $\boldsymbol{w}$, which motivates the need for linearization. Under Newton linearization, the nonlinear term can be approximated as

$$(uu_x)^{k+1} \approx u^{k+1}u_x^k + u^k u_x^{k+1} - u^k u_x^k,$$

where $k$ and $k+1$ denote the current and next nonlinear iteration step, respectively. Given initial guesses for $u$ and $u_x$ (e.g., setting $u$ and $u_x$ at all time steps equal to the initial field), the corresponding entries in $\mathbf{A}$ and $\mathbf{b}$ are updated as:

$$\mathcal{N}_\theta\big[f_j^{k+1}(\mathbf{x}_n^{\mathrm{pde}};\tilde{\boldsymbol{w}})\big] = \frac{\partial f_j^{k+1}(\mathbf{x}_n^{\mathrm{pde}};\tilde{\boldsymbol{w}})}{\partial t} + u^k(\mathbf{x}_n^{\mathrm{pde}})\frac{\partial f_j^{k+1}(\mathbf{x}_n^{\mathrm{pde}};\tilde{\boldsymbol{w}})}{\partial x}$$
$$+ f_j^{k+1}(\mathbf{x}_n^{\mathrm{pde}};\tilde{\boldsymbol{w}})\frac{\partial u^k(\mathbf{x}_n^{\mathrm{pde}})}{\partial x} - \frac{\partial^2 f_j^{k+1}(\mathbf{x}_n^{\mathrm{pde}};\tilde{\boldsymbol{w}})}{\partial x^2}, \tag{10}$$

$$q(\mathbf{x}_n^{\mathrm{pde}}) = u^k(\mathbf{x}_n^{\mathrm{pde}})\frac{\partial u^k(\mathbf{x}_n^{\mathrm{pde}})}{\partial x}. \tag{11}$$

This leads to an iterative nonlinear solve, where Eq. (9) is repeatedly solved to update $\boldsymbol{w}$ toward the optimal solution.

**How to make Tikhonov solver tractable?** The task adaptation is formulated as a Tikhonov-regularized least-squares problem, whose tractability mainly depends on the size of $\mathbf{A}$, the Tikhonov regularization parameter ($\lambda_{\mathrm{reg}}$), the PDE-loss importance hyperparameter ($\lambda_{\mathrm{pde}}$), and the choice of the initial field during nonlinear iterations.

- The matrix $\mathbf{A}$ is constructed from the network features at each sampling point of the PDE residuals/ICs/BCs, which are generated by a multilayer neural representation with skip connections. The distribution of these features directly determines the conditioning of the solve. In addition, when the system requires a large number of sampling points, the number of rows in $\mathbf{A}$ increases, making it more difficult to solve. Since the Tikhonov update only involves $\mathbf{A}^T\mathbf{A} \in \mathbb{R}^{D_o N_n(L-1) \times D_o N_n(L-1)}$, the computational cost mainly depends on the network feature dimension. Thus, **the practical scaling for large collocation sets may derive more from the fact that one can still control the cost of the solve through meta-learning a better (yet minimal) set of pre-final layer features.**

- To avoid laborious hyperparameter tuning, both $\lambda_{\mathrm{reg}}$ and $\lambda_{\mathrm{pde}}$ are meta-learned jointly with the network parameters.

- For nonlinear problems, an inappropriate initial field may slow convergence. Therefore, we use the initial condition as $u_{\mathrm{guess}}$ (setting $u_{\mathrm{guess}}$ at all time steps equal to the initial field) for time-dependent problems and employ temporal domain decomposition for more complex dynamics, ensuring that each block starts closer to the true solution, which improves the stability of the least-squares updates.

The specific discussion on the scalability of the model is provided in Appendix E.

### A.3. Meta-learning PINN on training tasks

The trainable hyperparameters during meta-learning include network weights ($\tilde{\boldsymbol{w}}$), PDE loss importance hyperparameter ($\lambda_{\text{pde}}$), and regularization parameter ($\lambda_{\text{reg}}$), collectively denoted as $\Theta$. These components are crucial for achieving optimal performance in task-specific physics-informed learning.

The meta-learning objective is to optimize $\Theta$ to learn task-agnostic representations spanning a family of PDEs and enable fast generalization to unseen tasks requiring only Tikhonov regularization update. In our framework, the outer loop of meta-learning updates the learnable parameters via gradient descent, while task-specific adaptation in the inner loop, or generalization, is performed through gradient-free Tikhonov regularization applied to the output layer weights. See Algorithm 1 for the pseudo-code workflow.

We consider a distribution over tasks $p(\mathcal{T})$ that we want our model to be able to adapt to. The meta-objective (loss) may consist solely of physics-based least-squares error ($l_{\text{LSE}}$, unsupervised), data-driven mean squared error ($l_{\text{MSE}}$, few-shot training), or a combination of both:

$$\min_{\Theta} \mathbb{E}_{\mathcal{T}_i \sim p(\mathcal{T})} \mathbb{E}_{\tilde{\boldsymbol{w}} \sim p_\theta(\tilde{\boldsymbol{w}})} \left[ l_{\text{LSE}}(\boldsymbol{w}^*) + l_{\text{MSE}}(\boldsymbol{w}^*) \right]. \tag{12}$$

Here, Tikhonov regularization is used to obtain the optimal output-layer weights $\boldsymbol{w}^*$ that allow the model to specialize to any realization of task $\mathcal{T}_i \sim p(\mathcal{T})$ for the given network's $\tilde{\boldsymbol{w}} \sim p_\theta(\tilde{\boldsymbol{w}})$.

$l_{\text{LSE}}$ and $l_{\text{MSE}}$ are

$$l_{\text{LSE}}(\boldsymbol{w}^*) = (\mathbf{A}\boldsymbol{w}^* - \mathbf{b})^T (\mathbf{A}\boldsymbol{w}^* - \mathbf{b}), \tag{13}$$

$$l_{\text{MSE}}(\boldsymbol{w}^*) = \frac{1}{n} \sum_{s=1}^{n} (u_s^{\text{label}} - \sum_j w_j^* f_j(x_s, t_s; \tilde{\boldsymbol{w}}))^2, \tag{14}$$

given labeled data $\{u_s^{\text{label}}\}_{s=1}^n$ ($n$ denotes the total number of collocation points including PDE residual, IC, and BC constraints for a task).

---

**Algorithm 1** Meta-learning on training tasks

    **Require:** Initialize $\Theta = (\tilde{\boldsymbol{w}}, \lambda_{\text{pde}}, \lambda_{\text{reg}})$; task distribution $p(\mathcal{T})$
    **Require:** Newton-linearized expressions for nonlinear terms
    **Require:** $\mathcal{C}$: constructor of least-squares system $(\mathbf{A}, \mathbf{b})$ (see Eq. (7))
  1: **while** not done **do**
  2:     Sample training tasks $\{\mathcal{T}_i\} \sim p(\mathcal{T})$
  3:     **for** each task $\mathcal{T}_i$ **do**
  4:         Initialize $u^{(1)}$
  5:         **for** $k = 1$ to N **do**         ▷ Nonlinear iterations with N steps
  6:             $(\mathbf{A}, \mathbf{b}) = \mathcal{C}(\mathcal{T}_i, \Theta, u^k)$
  7:             $\boldsymbol{w} = \begin{cases} (\lambda_{\text{reg}}\mathbf{I} + \mathbf{A}^T\mathbf{A})^{-1}\mathbf{A}^T\mathbf{b}, & \text{if tall} \\ \mathbf{A}^T(\lambda_{\text{reg}}\mathbf{I} + \mathbf{A}\mathbf{A}^T)^{-1}\mathbf{b}, & \text{if wide} \end{cases}$
  8:             $u^{(k+1)} = \boldsymbol{f}^T \boldsymbol{w}$
  9:         **end for**
10:         Compute loss $l_{\mathcal{T}_i}$ (see Eq. (13) and (14))
11:     **end for**
12:     $\Theta \leftarrow \Theta - \eta \nabla_\Theta \sum_i l_{\mathcal{T}_i}$
13: **end while**

---

We refer to $l_{\text{LSE}}$-based learning as **unsupervised learning**, and to $l_{\text{MSE}}$-based learning as **hybrid learning** (a combination of physics-based Tikhonov updates and data-driven loss). Unless otherwise stated, for meta-learning loss (outer-loop), our model does not consider the combined loss formulation $l_{\text{LSE}} + l_{\text{MSE}}$, as in some cases the combined objective exhibits poorer generalization performance compared with using $l_{\text{MSE}}$ alone.

In practice, we observe that for simple nonlinear PDEs (e.g., Burgers' equation), training with $l_{\text{LSE}}$ alone already leads to strong generalization. However, for more complex PDEs, incorporating a small amount of labeled data through the $l_{\text{MSE}}$ term becomes essential for both stability and accuracy.

---

**Algorithm 2** Gradient-free fine-tuning on new tasks

---

**Require:** Freeze $\Theta = (\tilde{w}, \lambda_{\text{pde}}, \lambda_{\text{reg}})$

1: **for** each test task $\mathcal{T}_i$ **do**
2:      Initialize $u^{(1)}$
3:      **for** $k = 1$ to N **do**
4:          $(\mathbf{A}, \mathbf{b}) = \mathcal{C}(\mathcal{T}_i, \Theta, u^k)$
5:          $\boldsymbol{w} = \begin{cases} (\lambda_{\text{reg}}\mathbf{I} + \mathbf{A}^T\mathbf{A})^{-1}\mathbf{A}^T\mathbf{b}, & \text{if tall} \\ \mathbf{A}^T(\lambda_{\text{reg}}\mathbf{I} + \mathbf{A}\mathbf{A}^T)^{-1}\mathbf{b}, & \text{if wide} \end{cases}$
6:          $u^{(k+1)} = \boldsymbol{f}^T\boldsymbol{w}$
7:      **end for**
8:      Store solution $u_{\mathcal{T}_i} = u^{(N+1)}$
9: **end for**
10: **return** all task solution $\{u_{\mathcal{T}_i}\}$

---

## A.4. Gradient-free fine-tuning to new tasks

After meta-learning, the network weights ($\tilde{\boldsymbol{w}}$), the PDE-loss importance hyperparameter ($\lambda_{\text{pde}}$), and the regularization parameter ($\lambda_{\text{reg}}$) are fixed. For a new task with different PDE parameters or IC/BC conditions, Tikhonov regularization is used to compute the output-layer weights ($\boldsymbol{w}$), enabling fast gradient-free adaptation to the new PDE instance. See Algorithm 2 for the pseudo-code workflow. Since this update is independent of gradient-based backpropagation optimizers such as stochastic gradient descent (SGD), the resulting adaptation is extremely fast while remaining physics-compliant.

## A.5. Linearization for nonlinear PDEs

It should be noted that the Tikhonov regularization is directly applicable to linear PDEs, where a linear matrix system can be constructed and solved in a single step. For nonlinear PDEs, the nonlinear terms must first be linearized so that the system can be cast into a linear form. The previous approaches typically relied on Picard method (lagging-of-coefficients), which linearizes the nonlinear terms using the solution from the previous iteration and then performs least-squares solves for multiple nonlinear iterations.

*Picard iterations.* Take the Burgers' equation as an example: $u_t + uu_x - \gamma u_{xx} = 0$. The Picard method approximately linearizes the nonlinear term as: $(uu_x)^{k+1} = u^k u_x^{k+1}$, where $k$ and $k + 1$ denote the current and next iteration steps, respectively. An initial guess for $u$ is provided, e.g., setting $u$ at all time steps equal to the initial field. Under this formulation, the entries of the least-squares matrix $\mathbf{A}$ and vector $\mathbf{b}$ in Eq. (7) corresponding to the $(k + 1)$-th iteration, $n$-th PDE collocation point, and the $j$-th output-layer neuron are given by:

$$\mathcal{N}_\theta\big[f_j^{k+1}(\mathbf{x}_n^{\text{pde}}; \tilde{\boldsymbol{w}})\big] = \frac{\partial f_j^{k+1}(\mathbf{x}_n^{\text{pde}}; \tilde{\boldsymbol{w}})}{\partial t} + u^k(\mathbf{x}_n^{\text{pde}})\frac{\partial f_j^{k+1}(\mathbf{x}_n^{\text{pde}}; \tilde{\boldsymbol{w}})}{\partial x}$$
$$- \frac{\partial^2 f_j^{k+1}(\mathbf{x}_n^{\text{pde}}; \tilde{\boldsymbol{w}})}{\partial x^2} \tag{15}$$

$$q(\mathbf{x}_n^{\text{pde}}) = 0, \tag{16}$$

where $f_j$ denotes the pre-final output of the neural network. Note that $u^k$ here is the known solution, either the initial guess or the solution at the $k$-th iteration.

The final output weights $\boldsymbol{w}^T = [\ldots, w_j, \ldots]$ are obtained via Tikhonov regularization-based nonlinear iterations. At each iteration, the solution is updated by $u(\mathbf{x}) = \sum_j w_j f_j$, which then serves as the new guess for the next iteration. This procedure is repeated until $\boldsymbol{w}$ satisfies a prescribed convergence criterion or the maximum number of iterations is reached.

*Newton linearization iterations.* The Picard approach suffers from slow convergence of nonlinear PDE solutions due to its linear convergence speed. To overcome these limitations, we use the Newton linearization approach.

Let $F(u^{k+1})$ denote the nonlinear term as a function of the state variable $u$. In Newton linearization, a first-order Taylor

expansion is performed around the current iterate $u^k$, yielding

$$F(u^{k+1}) \quad = F(u^k) + \frac{dF}{du}\Big|^k (u^{k+1} - u^k) + \text{H.O.T,} \tag{17}$$

where 'H.O.T" denotes the higher-order truncated terms.

According to Eq. (17), we can derive a general expression for the nonlinear terms $u^m$ with $m$ denoting the exponent that commonly appear in PDEs, as follows:

$$\begin{aligned} (u^m)^{k+1} &\approx (u^m)^k + m(u^{m-1})^k(u^{k+1} - u^k) \\ &= m(u^k)^{m-1}u^{k+1} + (1-m)(u^k)^m \end{aligned} \tag{18}$$

The corresponding expressions for $m = 2$ and $m = 3$ are as follows:

$$(u^2)^{k+1} \quad \approx \quad 2u^{k+1}u^k - u^k u^k \tag{19}$$

$$(u^3)^{k+1} \quad \approx \quad 3u^{k+1}u^k u^k - 2u^k u^k u^k \tag{20}$$

For nonlinear terms involving multiple state variables, i.e., $F(u^{k+1}, v^{k+1})$, the corresponding Taylor expansion is given by:

$$F(u^{k+1}, v^{k+1}) = F(u^k, v^k) + \frac{\partial F}{\partial u}\Big|^k (u^{k+1} - u^k) + \frac{\partial F}{\partial v}\Big|^k (v^{k+1} - v^k) + \text{H.O.T} \tag{21}$$

Based on Eq. (21), the general Newton linearization form for nonlinear convective terms of arbitrary order can be expressed as:

$$\begin{aligned} (u^m u_x)^{k+1} &\approx (u^m)^k u_x^k + u_x^k \left[ (u^m)^{k+1} - (u^m)^k \right] + (u^m)^k (u_x^{k+1} - u_x^k) \\ &= (u^m)^{k+1} u_x^k + (u^m)^k u_x^{k+1} - (u^m)^k u_x^k \\ &= \left[ m(u^k)^{m-1} u^{k+1} + (1-m)(u^k)^m \right] u_x^k + (u^k)^m u_x^{k+1} - (u^k)^m u_x^k \\ &= m(u^k)^{m-1} u_x^k u^{k+1} + (1-m)(u^k)^m u_x^k + (u^k)^m u_x^{k+1} - (u^k)^m u_x^k \\ &= m(u^k)^{m-1} u_x^k u^{k+1} + (u^k)^m u_x^{k+1} - m(u^k)^m u_x^k \end{aligned} \tag{22}$$

Accordingly, the corresponding expressions for the nonlinear convective terms with $m = 1$, $m = 2$, and $m = 3$ are given by:

$$(uu_x)^{k+1} \quad \approx \quad u^{k+1}u_x^k + u^k u_x^{k+1} - u^k u_x^k \tag{23}$$

$$(u^2 u_x)^{k+1} \quad \approx \quad 2u^k u_x^k u^{k+1} + (u^k)^2 u_x^{k+1} - 2(u^k)^2 u_x^k \tag{24}$$

$$(u^3 u_x)^{k+1} \quad \approx \quad 3(u^k)^2 u_x^k u^{k+1} + (u^k)^3 u_x^{k+1} - 3(u^k)^3 u_x^k \tag{25}$$

Therefore, the nonlinear term in the Burgers' equation under Newton linearization can be expressed as $(uu_x)^{k+1} \approx \underline{u^{k+1}u_x^k} + u^k u_x^{k+1} - \underline{u^k u_x^k}$, where the underlined terms originate from higher-order corrections that refine the classical Picard linearization: $(uu_x)^{k+1} = u^k u_x^{k+1}$. Given initial guesses for $u$ and $u_x$ (e.g., setting $u$ and $u_x$ at all time steps equal to the initial field), the corresponding entries in **A** and **b** are updated as:

$$\begin{aligned} \mathcal{N}_\theta \left[ f_j^{k+1}(\mathbf{x}_n^{\text{pde}}; \tilde{\boldsymbol{w}}) \right] = {} & \frac{\partial f_j^{k+1}(\mathbf{x}_n^{\text{pde}}; \tilde{\boldsymbol{w}})}{\partial t} + u^k(\mathbf{x}_n^{\text{pde}}) \frac{\partial f_j^{k+1}(\mathbf{x}_n^{\text{pde}}; \tilde{\boldsymbol{w}})}{\partial x} \\ & + f_j^{k+1}(\mathbf{x}_n^{\text{pde}}; \tilde{\boldsymbol{w}}) \frac{\partial u^k(\mathbf{x}_n^{\text{pde}})}{\partial x} - \frac{\partial^2 f_j^{k+1}(\mathbf{x}_n^{\text{pde}}; \tilde{\boldsymbol{w}})}{\partial x^2} \end{aligned} \tag{26}$$

$$q(\mathbf{x}_n^{\text{pde}}) = u^k(\mathbf{x}_n^{\text{pde}}) \frac{\partial u^k(\mathbf{x}_n^{\text{pde}})}{\partial x}. \tag{27}$$

The Newton linearization method substantially accelerates the convergence of Tikhonov regularization-based nonlinear iterations due to its quadratic convergence property (see Appendix B for a proof). Consequently, it reduces the time complexity of meta-learning and enhances the speed of fine-tuning and generalization on new tasks.

Similarly, the Newton-linearized expressions for other nonlinear terms, such as hyperbolic, exponential, and logarithmic forms, are given as follows:

$$
\begin{aligned}
[\sinh(u)]^{k+1} &\approx \sinh(u^k) + \cosh(u^k)(u^{k+1} - u^k) \\
&= \cosh(u^k)u^{k+1} + \sinh(u^k) - \cosh(u^k)u^k
\end{aligned}
\tag{28}
$$

$$
\begin{aligned}
[\cosh(u)]^{k+1} &\approx \cosh(u^k) + \sinh(u^k)(u^{k+1} - u^k) \\
&= \sinh(u^k)u^{k+1} + \cosh(u^k) - \sinh(u^k)u^k
\end{aligned}
\tag{29}
$$

$$
\begin{aligned}
[\tanh(u)]^{k+1} &\approx \tanh(u^k) + \left(1 - \tanh^2(u^k)\right)(u^{k+1} - u^k) \\
&= \left(1 - \tanh^2(u^k)\right)u^{k+1} + \tanh(u^k) \\
&\quad - \left(1 - \tanh^2(u^k)\right)u^k
\end{aligned}
\tag{30}
$$

$$
\begin{aligned}
[\exp(u)]^{k+1} &\approx \exp(u^k) + \exp(u^k)(u^{k+1} - u^k) \\
&= \exp(u^k)u^{k+1} + \exp(u^k) - \exp(u^k)u^k
\end{aligned}
\tag{31}
$$

$$
\begin{aligned}
[u\ln(u)]^{k+1} &\approx u^k\ln(u^k) + (\ln(u^k) + 1)(u^{k+1} - u^k) \\
&= (\ln(u^k) + 1)u^{k+1} + u^k\ln(u^k) - (\ln(u^k) + 1)u^k
\end{aligned}
\tag{32}
$$

## B. Convergence analysis of the Newton linearization

Newton linearization has long been employed in numerical PDE solvers (Sheu & Lin, 2004; 2005; Chiu et al., 2008) (e.g., to linearize the convective term in the Navier-Stokes equations), and it is well known to achieve quadratic convergence, in contrast to the linear convergence of Picard iteration (Dennis Jr & Schnabel, 1996). Our contribution is that we innovatively integrate the Newton linearization into the Tikhonov-regularized PINN, which significantly enhances the performance for meta-learning nonlinear PDEs. This appendix provides a theoretical analysis showing that the Newton linearization used in the Tikhonov-regularized PINN is essentially equivalent to the classical Newton method, and therefore achieves the quadratic convergence guarantees.

### B.1. Nonlinear system formulation

The output $u$ and its spatial derivative $u_x$ at the $n$-th collocation point in the Tikhonov-regularized PINN are given by

$$
u(\mathbf{x}_n) = \sum_j f_j(\mathbf{x}_n)w_j, \quad u_x(\mathbf{x}_n) = \sum_j f_{x,j}(\mathbf{x}_n)w_j \quad \Rightarrow \quad u = \mathbf{\Phi}\boldsymbol{w}, \quad u_x = \mathbf{\Phi}_x\boldsymbol{w},
\tag{33}
$$

where the pre-final features $\{f_j\}$ are generated by hidden-layer transformations followed by smooth activation functions, and are thus inherently bounded and continuously differentiable. Here, $\boldsymbol{w}^T = [\ldots, w_j, \ldots]$ denotes the output-layer weights, $\Phi_{nj} = f_j(\mathbf{x}_n)$, and $(\Phi_x)_{nj} = f_{x,j}(\mathbf{x}_n)$.

To provide a simple illustration, we consider a nonlinear equation $F = uu_x$ defined on the domain $x \in [0, 1]$, subject to the boundary conditions $u(0) = 0$ and $u(1) = 1$. The PDE residuals evaluated at the collocation points $\{x_n\}$ are then given by:

$$
r_n(\boldsymbol{w}) = u(x_n)\, u_x(x_n) = (\mathbf{\Phi}\boldsymbol{w})_n\, (\mathbf{\Phi}_x\boldsymbol{w})_n.
\tag{34}
$$

The residuals corresponding to the boundary conditions are

$$r_0(\boldsymbol{w}) = u(0) - 0 = \boldsymbol{\Phi}(0,:)\boldsymbol{w}, \qquad r_1(\boldsymbol{w}) = u(1) - 1 = \boldsymbol{\Phi}(1,:)\boldsymbol{w} - 1. \tag{35}$$

We consider the nonlinear system of equations in a general form: $\mathbf{A}(\boldsymbol{w})\,\boldsymbol{w} = \mathbf{b}$.

By stacking all the residuals, we obtain the unified nonlinear system:

$$\begin{aligned}
F(\boldsymbol{w}) &= \mathbf{A}(\boldsymbol{w})\,\boldsymbol{w} - \mathbf{b} \\
&= \begin{bmatrix} u \odot u_x \\ \boldsymbol{\Phi}(0,:)\boldsymbol{w} \\ \boldsymbol{\Phi}(1,:)\boldsymbol{w} - 1 \end{bmatrix} = \begin{bmatrix} (\boldsymbol{\Phi}\boldsymbol{w}) \odot (\boldsymbol{\Phi}_x\boldsymbol{w}) \\ \boldsymbol{\Phi}(0,:)\boldsymbol{w} \\ \boldsymbol{\Phi}(1,:)\boldsymbol{w} - 1 \end{bmatrix} = \mathbf{0},
\end{aligned} \tag{36}$$

where $\odot$ denotes the elementwise product.

### B.2. Classical Newton method

For the nonlinear system $F(\boldsymbol{w}) = 0$, the Newton method applies a first-order Taylor expansion around the current iterate $\boldsymbol{w}^k$ (Dennis Jr & Schnabel, 1996):

$$F(\boldsymbol{w}^{k+1}) \approx F(\boldsymbol{w}^k) + J_F(\boldsymbol{w}^k)\,(\boldsymbol{w}^{k+1} - \boldsymbol{w}^k) = 0. \tag{37}$$

Solving for the update $\Delta\boldsymbol{w} = \boldsymbol{w}^{k+1} - \boldsymbol{w}^k$ leads to the following linear system:

$$J_F(\boldsymbol{w}^k)\,(\boldsymbol{w}^{k+1} - \boldsymbol{w}^k) = -F(\boldsymbol{w}^k), \tag{38}$$

which can be equivalently reformulated as:

$$\boxed{J_F(\boldsymbol{w}^k)\,\boldsymbol{w}^{k+1} = J_F(\boldsymbol{w}^k)\,\boldsymbol{w}^k - F(\boldsymbol{w}^k).} \tag{39}$$

### B.3. Our Newton-PINet: Newton linearization used in Tikhonov-regularized PINN

In our model, the nonlinear terms are first linearized such that the resulting system can be written in the linear form of Eq. (7), which leads to an iterative nonlinear procedure to update $\boldsymbol{w}$ toward the optimal solution. After linearization, each nonlinear iteration requires solving a linear least-squares system of the form:

$$\mathbf{A}(\boldsymbol{w}^k)\,\boldsymbol{w}^{k+1} = \mathbf{b}^k. \tag{40}$$

Using the Newton expansion for $uu_x$, as shown in Eq. (23), the linearization of $(uu_x)^{k+1}$ can be written as:

$$(uu_x)^{k+1} \approx u^{k+1}u_x^k + u^k u_x^{k+1} - u^k u_x^k, \tag{41}$$

where $k$ and $k+1$ denote the current and next iteration step, respectively. We define $u^k = \boldsymbol{\Phi}\boldsymbol{w}^k$, $u_x^k = \boldsymbol{\Phi}_x\boldsymbol{w}^k$. Then we have

$$\begin{aligned}
(uu_x)^{k+1} &= (\boldsymbol{\Phi}\boldsymbol{w})^{(k+1)} \odot (\boldsymbol{\Phi}_x\boldsymbol{w})^{(k+1)} \\
&\approx (\boldsymbol{\Phi}_x\boldsymbol{w}^k) \odot (\boldsymbol{\Phi}\boldsymbol{w}^{(k+1)}) + (\boldsymbol{\Phi}\boldsymbol{w}^k) \odot (\boldsymbol{\Phi}_x\boldsymbol{w}^{(k+1)}) - (\boldsymbol{\Phi}\boldsymbol{w}^k) \odot (\boldsymbol{\Phi}_x\boldsymbol{w}^k).
\end{aligned} \tag{42}$$

Through matrix manipulations, we obtain

$$\begin{aligned}
(\boldsymbol{\Phi}_x\boldsymbol{w}^k) \odot (\boldsymbol{\Phi}\boldsymbol{w}^{(k+1)}) &= \mathrm{diag}(\boldsymbol{\Phi}_x\boldsymbol{w}^k)(\boldsymbol{\Phi}\boldsymbol{w}^{(k+1)}) = \big(\mathrm{diag}(\boldsymbol{\Phi}_x\boldsymbol{w}^k)\boldsymbol{\Phi}\big)\boldsymbol{w}^{(k+1)}, \\
(\boldsymbol{\Phi}\boldsymbol{w}^k) \odot (\boldsymbol{\Phi}_x\boldsymbol{w}^{(k+1)}) &= \mathrm{diag}(\boldsymbol{\Phi}\boldsymbol{w}^k)(\boldsymbol{\Phi}_x\boldsymbol{w}^{(k+1)}) = \big(\mathrm{diag}(\boldsymbol{\Phi}\boldsymbol{w}^k)\boldsymbol{\Phi}_x\big)\boldsymbol{w}^{(k+1)}.
\end{aligned} \tag{43}$$

Consequently, in the system $\mathbf{A}(\boldsymbol{w}^k)\,\boldsymbol{w}^{k+1} = \mathbf{b}^k$, the matrix and right-hand side are given by

$$\mathbf{A}(\boldsymbol{w}^k) = \begin{bmatrix} \big(\mathrm{diag}(\boldsymbol{\Phi}_x\boldsymbol{w}^k)\boldsymbol{\Phi}\big) + \big(\mathrm{diag}(\boldsymbol{\Phi}\boldsymbol{w}^k)\boldsymbol{\Phi}_x\big) \\ \boldsymbol{\Phi}(0,:) \\ \boldsymbol{\Phi}(1,:) \end{bmatrix},$$

$$\mathbf{b}^k = \begin{bmatrix} (\mathbf{\Phi}\boldsymbol{w}^k) \odot (\mathbf{\Phi}_x \boldsymbol{w}^k) \\ 0 \\ 1 \end{bmatrix}. \tag{44}$$

It can be verified that our $\mathbf{A}(\boldsymbol{w}^k)$ and $\mathbf{b}^k$ correspond exactly to the classical Newton method:

$$\boxed{J_F(\boldsymbol{w}^k)\,\boldsymbol{w}^{k+1} = J_F(\boldsymbol{w}^k)\boldsymbol{w}^k - F(\boldsymbol{w}^k),}$$

i.e.,

$$\mathbf{A}(\boldsymbol{w}^k) = J_F(\boldsymbol{w}^k), \tag{45}$$

$$\begin{aligned}
\mathbf{b}^k &= J_F(\boldsymbol{w}^k)\,\boldsymbol{w}^k - F(\boldsymbol{w}^k) \\
&= \begin{bmatrix} (\mathbf{\Phi}\boldsymbol{w}^k) \odot (\mathbf{\Phi}_x \boldsymbol{w}^k) + (\mathbf{\Phi}\boldsymbol{w}^k) \odot (\mathbf{\Phi}_x \boldsymbol{w}^k) - (\mathbf{\Phi}\boldsymbol{w}^k) \odot (\mathbf{\Phi}_x \boldsymbol{w}^k) \\ 0 \\ 1 \end{bmatrix} \\
&= \begin{bmatrix} (\mathbf{\Phi}\boldsymbol{w}^k) \odot (\mathbf{\Phi}_x \boldsymbol{w}^k) \\ 0 \\ 1 \end{bmatrix}.
\end{aligned} \tag{46}$$

Therefore, the Newton linearization employed in the Tikhonov-regularized PINN retains the same quadratic convergence guarantees as the classical Newton method.

### B.4. PINet: Pichard linearization used in Tikhonov-regularized PINN

As a baseline model, we employ Picard linearization in the PINet framework. For the Picard linearization, also referred to as the "lagging-of-coefficients" approach, one factor in the nonlinear term $uu_x$ is kept from the previous iteration $k$. Specifically, the linearization of $(uu_x)^{k+1}$ is given by $(uu_x)^{k+1} \approx u^k\,u_x^{k+1}$, which can be written in matrix form as

$$\begin{aligned}
(uu_x)^{k+1} &\approx (\mathbf{\Phi}\boldsymbol{w})^k \odot (\mathbf{\Phi}_x \boldsymbol{w})^{(k+1)} \\
&= \big(\operatorname{diag}(\mathbf{\Phi}\boldsymbol{w}^k)\mathbf{\Phi}_x\big)\boldsymbol{w}^{(k+1)}.
\end{aligned} \tag{47}$$

Consequently, in the linear system $\mathbf{A}(\boldsymbol{w}^k)\,\boldsymbol{w}^{k+1} = \mathbf{b}^k$, the corresponding matrix and right-hand side are

$$\mathbf{A}(\boldsymbol{w}^k) = \begin{bmatrix} \big(\operatorname{diag}(\mathbf{\Phi}\boldsymbol{w}^k)\mathbf{\Phi}_x\big) \\ \mathbf{\Phi}(0,:) \\ \mathbf{\Phi}(1,:) \end{bmatrix}, \quad \mathbf{b}^k = \begin{bmatrix} 0 \\ 0 \\ 1 \end{bmatrix}. \tag{48}$$

Since the Picard method only updates one factor at each iteration while keeping the other fixed, the convergence rate is generally linear rather than quadratic.

## C. Problem description and experimental results

### C.1. Description of nonlinear PDE problems

### (i) Nonlinear convection-type PDEs

Burgers' equation (1D + time)

$$\frac{\partial u}{\partial t} + u\frac{\partial u}{\partial x} - \gamma\frac{\partial^2 u}{\partial x^2} = 0, \quad x \in [-1, 1],\ t \in [0, 1].$$

a) Varying PDE parameter:

The temporal and spatial resolutions are set to $51 \times 129$ with periodic boundary conditions. We fix the initial condition as $u(x,0) = -\sin(\pi x)$, and vary the viscosity parameter $\gamma$ within the range $[0.001, 0.05]$ with an increment of $0.001$ to generate 50 tasks; 16 of them are randomly selected as training tasks, while the remaining 34 are used for testing. The model architecture consists of 4 hidden layers with 128 neurons per layer. Time block decomposition is not employed (a single time block), and the step of the fine-tuning (nonlinear) iterations is set to 6.

b) Varying initial condition:

The temporal and spatial resolutions are set to $101 \times 101$ with periodic boundary conditions. Four distinct viscosity values $\gamma \in \{0.1, 0.01, 0.001, 0.0001\}$ are considered. For each $\gamma$, the initial conditions are sampled from a Gaussian random field $\mathcal{N}(0, 625(-\Delta + 25I)^{-4})$, consistent with (Wang et al., 2021). We generate 50 training tasks and 1000 test tasks for each viscosity, and experiments are conducted independently for each $\gamma$. The model architecture consists of 4 hidden layers with 128 neurons per layer, and the nonlinear iteration step is set to 4. We evaluate two model variants: unsupervised Newton-PINet (1 time block) and hybrid Newton-PINet (4 time blocks).

Generalized Korteweg-de Vries (KdV) equation (1D + time)

$$\frac{\partial u}{\partial t} + \beta_1 u \frac{\partial u}{\partial x} + \beta_2 u^2 \frac{\partial u}{\partial x} + \beta_3 u^3 \frac{\partial u}{\partial x} + \delta \frac{\partial^3 u}{\partial x^3} = 0, \quad x \in [-1, 1],\ t \in [0, 1]$$

The temporal and spatial resolutions are set to $101 \times 257$ with periodic boundary conditions. We consider three different forms of nonlinear convection terms, corresponding to three combinations of $(\beta_1, \beta_2, \beta_3)$. For each combination, 15 tasks are randomly generated by varying $\delta$, leading to a total of 45 tasks. Specifically:

$$\text{KdV: } (\beta_1, \beta_2, \beta_3) = (1, 0, 0), \quad \delta \in [0.032^2, 0.12^2],$$

$$M\text{-KdV: } (\beta_1, \beta_2, \beta_3) = (0, 1, 0), \quad \delta \in [0.02^2, 0.06^2],$$

$$G\text{-KdV: } (\beta_1, \beta_2, \beta_3) = (0, 0, 1), \quad \delta \in [0.08^2, 0.15^2].$$

Among them, 10 tasks are randomly chosen for training and the remaining 35 tasks for testing. The model architecture consists of 4 hidden layers with 256 neurons per layer. 6 time blocks are employed, and the nonlinear iteration step is set to 6.

Kuramoto-Sivashinsky (K-S) equation (1D + time)

$$\frac{\partial u}{\partial t} + \beta u \frac{\partial u}{\partial x} - \gamma \frac{\partial^2 u}{\partial x^2} + \sigma \frac{\partial^4 u}{\partial x^4} = 0, \quad x \in [0, 2\pi],\ t \in [0, 1]$$

The simulated data have a temporal and spatial resolution of $251 \times 509$, which is downsampled to $63 \times 129$ for training and evaluation. Periodic boundary conditions are imposed. The fixed coefficients are set as $\beta = 100/16$ and $\gamma = 100/16^2$. The parameter $\sigma$ varies within the range $[200/16^4, 300/16^4]$, along with varying initial conditions:

$$u(x, 0) = \sum_{j=1}^{J} A_j \sin\left(\frac{2\pi l_j x}{L} + \phi_j\right),$$

where $J = 5$, $L = 1$, $A_j \in [-0.8, 0.8]$, $l_j \in \{0, 1, 2, 3, 4\}$, and $\phi_j \in [-\pi, \pi]$. In total, 150 tasks are generated, of which 100 are used for training and 50 for testing. The model architecture consists of 4 hidden layers with 256 neurons per layer. 15 time blocks are employed, and the nonlinear iteration step is set to 4.

Lid-driven cavity (LDC) equations (2D)

The 2D steady incompressible Navier-Stokes equations in LDC problem are given by

$$\begin{cases} \frac{\partial u}{\partial x} + \frac{\partial v}{\partial y} = 0, \\[2mm] u \frac{\partial u}{\partial x} + v \frac{\partial u}{\partial y} + \frac{\partial p}{\partial x} - \frac{1}{Re}\left(\frac{\partial^2 u}{\partial x^2} + \frac{\partial^2 u}{\partial y^2}\right) = 0, \\[2mm] u \frac{\partial v}{\partial x} + v \frac{\partial v}{\partial y} + \frac{\partial p}{\partial y} - \frac{1}{Re}\left(\frac{\partial^2 v}{\partial x^2} + \frac{\partial^2 v}{\partial y^2}\right) = 0, \end{cases}$$

where $(x, y) \in [0, 1]^2$ and $Re$ is the Reynolds number.

The simulated data are generated on a $200 \times 200$ grid and then downsampled to $51 \times 51$ for training and testing. The boundary conditions are

$$\text{top lid: } u = 1, v = 0; \quad \text{other walls: } u = v = 0.$$

The $Re$ ranges from 1 to 1000 (specifically 1, 5, 10, 25, 50, 80, 100, ..., 1000 with a step of 50 from 100 to 1000), yielding 25 flow conditions. We use $Re$ = 1, 50, 200, 400, 600, 800 as the training set and the remaining cases as the test set. The model consists of 4 hidden layers with 128 neurons per layer. With three output variables $(u, v, p)$, the resulting weight dimension of the output layer (i.e., the number of columns in matrix $\mathbf{A}$) is $128 \times 4 \times 3 = 1536$. The nonlinear iteration step is set to 8.

**(ii) Nonlinear forcing-type PDEs**

Convection-diffusion-reaction (CDR) equation (1D + time)

$$\frac{\partial u}{\partial t} + \alpha \frac{\partial u}{\partial x} - \gamma \frac{\partial^2 u}{\partial x^2} - \rho_1 u + \rho_2 u^2 + \rho_3 u^3 = 0, \quad x \in [0, 1], \, t \in [0, 1].$$

The simulated data have a temporal and spatial resolution of $201 \times 257$, which are downsampled to $101 \times 129$ for training and evaluation. Periodic boundary conditions are imposed. The parameters are set as $\alpha = 1$, $\gamma \in \{0.005, 0.01, 0.05\}$, $\rho_1, \rho_2 \in \{0, 1, 3, 5\}$, and $\rho_3 = 5$. The initial conditions are generated from

$$u(x, 0) = \sum_{j=1}^{J} A_j \sin(l_j x + \phi_j),$$

where $J = 5$, $A_j \in [0.1, 0.5]$, $l_j \in \{1, 2, 3, 4\}$, and $\phi_j \in [-\pi, \pi]$.

In total, 200 tasks are generated, with 40 used for training and 160 for testing. The model architecture consists of 4 hidden layers with 128 neurons per layer. 4 time blocks are employed, and the nonlinear iteration step is set to 4.

Klein-Gordon equation (1D + time)

$$\frac{\partial^2 u}{\partial t^2} - \gamma \frac{\partial^2 u}{\partial x^2} + u^3 = q, \quad x \in [0, 1], \, t \in [0, 1]$$

$$\text{ICs:} \quad u(x, 0) = 0, \quad \frac{\partial u}{\partial t}(x, 0) = 0,$$

where the solution is defined as $u(x, t) = k_1 x \cos(k_2 \pi t) + k_3 (xt)^3$, which is used to derive the corresponding Dirichlet BCs, and source term $q$. The spatio-temporal grid resolution is set to $32 \times 32$ for meta-learning, while a finer grid of $128 \times 128$ is used for testing. The parameters vary within $\gamma, k_1, k_3 \in [0.5, 3]$ and $k_2 \in [1, 7]$. A total of 80 tasks are generated, with 16 used for meta-learning and 64 for testing. The model network consists of 4 hidden layers with 128 neurons per layer. 1 time block is employed, and the nonlinear iteration step is set to 4.

Hyperbolic heat equation (1D + time)

$$\frac{\partial u}{\partial t} - \gamma \frac{\partial^2 u}{\partial x^2} + k_1 \sinh(u) + k_2 \cosh(u) + k_3 \tanh(u) = q, \quad x \in [-1, 1], \, t \in [0, 1],$$

where the solution is defined as $u(x, t, k_1, k_2, k_3) = \sin(k_1 k_2 \pi x) \cos(k_3 \pi x) e^{-\pi t^2}$, which is used to derive the corresponding ICs, Dirichlet BCs, and source term $q$. The spatio-temporal grid resolution is set to $32 \times 32$ for meta-learning, while a finer grid of $128 \times 128$ is used for testing. The parameters vary within $\gamma \in [0.2, 3]$ and $k_1, k_2, k_3 \in [0.2, 2.5]$. A total of 80 tasks are generated, with 16 used for training and 64 for testing. The model network consists of 2 hidden layers with 450 neurons per layer. 1 time block is employed, and the nonlinear iteration step is set to 4.

Logarithmic heat equation (1D + time)

$$\frac{\partial u}{\partial t} - \gamma \frac{\partial^2 u}{\partial x^2} + k_1 u \log(k_2 u) + k_3 e^{k_4 u} = q, \quad x \in [-1, 1], \, t \in [0, 1],$$

where the solution is defined as $u(x, t, k_2, k_4) = \big(\sin(k_2\pi x) + 1.5\big)e^{-k_4\pi x^2}e^{-\pi t^2}$, which is used to derive the corresponding ICs, Dirichlet BCs, and source term $q$. The spatio-temporal grid resolution is set to $32 \times 32$ for meta-learning, while a finer grid of $128 \times 128$ for testing. The parameters vary within $\gamma, k_1, k_3 \in [0.5, 5]$ and $k_2, k_4 \in [0.5, 2]$. A total of 80 tasks are generated, with 16 used for meta-learning and 64 for testing. The model network consists of 2 hidden layers with 450 neurons per layer. 1 time block is employed, and the nonlinear iteration step is set to 4.

Helmholtz equation (2D)

$$\frac{\partial^2 u}{\partial x^2} + \frac{\partial^2 u}{\partial y^2} - 100u + 10\cosh(u) = q, \quad x, y \in [0, 1.5],$$

where the solution is defined as $u(x, y) = 4\cos(\alpha_1\pi x^2)\cos(\alpha_2\pi y^2)$, which is used to derive the corresponding Dirichlet BCs, and source term $q$. The spatial grid resolution is set to $32 \times 32$ for meta-learning, while a finer grid of $128 \times 128$ is used for testing. The parameters vary within $\alpha_1, \alpha_2 \in [1, 3]$. A total of 80 tasks are generated, with 20 used for meta-learning and 60 for testing. The model network consists of 2 hidden layers with 450 neurons per layer. The nonlinear iteration step is set to 4.

Parametric diffusion-reaction equation (2D)

$$\frac{\partial^2 u}{\partial x^2} + \frac{\partial^2 u}{\partial y^2} + u(1 - u^2) = q, \quad x, y \in [-1, 1],$$

where the solution is defined as $u(x, y, a_1, a_2, w_1, w_2, w_3, w_4) = a_1\tanh(w_1 x)\tanh(w_2 y) + a_2\sin(w_3 x)\sin(w_4 y)$ is used to derive the corresponding Dirichlet BCs and source term $q$. The spatial grid resolution is set to $32 \times 32$ for meta-learning, while a finer grid of $128 \times 128$ is used for testing. The parameters vary within $a_1, a_2 \in [0.1, 1]$ and $w_1, w_2, w_3, w_4 \in [1, 5]$. 17 tasks are generated for meta-learning and 100 for testing. The model network consists of 2 hidden layers with 450 neurons per layer. The nonlinear iteration step is set to 4.

## C.2. Data generation, error metrics, and computational setup

Datasets for Burgers, generalized KdV, K-S, and CDR problems are generated using the `Chebfun` package (Driscoll et al., 2014) with spectral Fourier discretization and a fourth-order exponential time-differencing Runge-Kutta scheme (ETDRK4) (Cox & Matthews, 2002). Datasets for LDC problem are generated using a computational fluid dynamics (CFD) method (Chiu & Poh, 2021). For the remaining benchmark problems, analytical solutions are available.

We evaluate the generalization accuracy of the models using both mean squared error (MSE) and relative $L^2$ error. In addition, for the 1D nonlinear diffusion-reaction problem, we adopt the relative MSE metric, consistent with the baseline study (Boudec et al., 2024) for comparison (see Table 3 and Fig. 10).

$$\text{MSE} = \frac{1}{n}\sum_{s=1}^{n}\left(u_s^{\text{label}} - u_s\right)^2, \tag{49}$$

$$\text{Relative } L^2 = \frac{\left\|u^{\text{label}} - u\right\|_2}{\left\|u^{\text{label}}\right\|_2}, \tag{50}$$

$$\text{Relative MSE} = \frac{\sum_{s=1}^{n}\left(u_s^{\text{label}} - u_s\right)^2}{\sum_{s=1}^{n}\left(u_s^{\text{label}}\right)^2}, \tag{51}$$

where $u^{\text{label}}$ and $u$ denote the ground-truth and predicted solutions for a given PDE instance, and $n$ is the total number of collocation points, including those for the PDE residual, IC/BC constraints.

In this study, all models are implemented in the JAX framework (Bradbury et al., 2018). For the KdV, K-S, CDR, and LDC problems, model training is parallelized across four Tesla V100 GPUs; all other models are trained on a single Tesla V100 GPU. Inference for all models is performed on a single Tesla V100 GPU.

## C.3. Summary of model performance on nonlinear PDE problems

Table 4 summarizes the meta-learning parameter configurations on nonlinear PDEs considered in this study, and Table 5 summarizes the model performance on each problem. For each problem, besides the **Newton-PINet** model proposed in

this study, two additional baseline models were tested: one that uses the Picard method, referred to as **PINet**, and a purely data-driven **DNN** baseline which incorporates PDE parameters as inputs. The reported meta-learning and fine-tuning time correspond to the Newton-PINet model. Note that this table does not include comparisons with other benchmark models on problems such as the Burgers' equation with varying initial conditions; the detailed results can be found in Table 2, Table 3, and Table 6.

As shown, Newton-PINet achieves superior test accuracy across nearly all nonlinear PDEs compared to DNN and PINet, except for the Klein-Gordon equation where PINet exhibits slightly better accuracy. Notably, for hyperbolic and logarithmic heat problems, the Picard approach employed by PINet fails to converge (leading to NaN values during meta-learning), whereas Newton-PINet remains stable and achieves accurate generalization.

*Table 4.* Summary of meta-learning parameter configurations on nonlinear PDEs in this study. For time-dependent PDEs, the mesh size is reported as $t \times x$, whereas for time-independent PDEs it is reported as $x \times y$. "Mesh size" and "No. **A** rows" refer to settings in the meta-learning stage unless otherwise marked with "(test)", which denotes the inference stage. "Dim." denotes the dimension, and "No." stands for "number of". "–" indicates that the time-blocking strategy is not applicable for time-independent problems.

| Problem | Mesh size (test) | Dim. $n_{pde} + n_{ic} + n_{bc}$; No. **A** rows (test) | Dim. $\boldsymbol{w}$; No. **A** columns | No. train/ test tasks | Nonlinear iteration | No. time blocks |
|---|---|---|---|---|---|---|
| Burgers' (1D + time) | 51×129 | 6810 | 512 | 16/34 | 6 | 1 |
| Generalized KdV (1D + time) | 101×257 | 4402 | 1024 | 10/35 | 6 | 6 |
| Kuramoto-Sivashinsky (1D + time) | 63×129 | 649 | 1024 | 100/50 | 4 | 15 |
| Convection-diffusion-reaction (1D + time) | 101×129 | 3404 | 512 | 40/160 | 4 | 4 |
| Klein-Gordon (1D + time) | 32×32 (128×128) | 1148 (16892) | 512 | 16/64 | 4 | 1 |
| Hyperbolic heat (1D + time) | 32×32 (128×128) | 1148 (16892) | 900 | 16/64 | 4 | 1 |
| Logarithmic heat (1D + time) | 32×32 (128×128) | 1148 (16892) | 900 | 16/64 | 4 | 1 |
| Helmholtz (2D) | 32×32 (128×128) | 1148 (16892) | 900 | 20/60 | 4 | – |
| Parametric diffusion-reaction (2D) | 32×32 (128×128) | 1148 (16892) | 900 | 17/100 | 4 | – |
| Lid-driven cavity (2D) | 51×51 | 7595 | 1536 | 6/19 | 8 | – |

**Prediction results of generalized KdV, K-S, and LDC problems:** Due to the coexistence of nonlinear convection terms and high-order dispersion operators, solving the generalized KdV and K-S equations is particularly challenging for PINN-based models. To mitigate these difficulties, we adopt a larger number of temporal domain partitions to stabilize training and improve convergence (6 time blocks for the generalized KdV and 15 time blocks for the K-S equations). As shown in Table 5 and Fig. 4, Newton-PINet achieves test errors of MSE $= 1.94 \times 10^{-3}$ on the generalized KdV, MSE $= 3.45 \times 10^{-2}$ on the K-S, and MSE $= 1.67 \times 10^{-4}$ on the LDC problem, outperforming PINet and DNN under the same configuration. Figures 5, 6, 7 show the prediction results of the Newton-PINet model on test tasks for the generalized KdV, K-S, and LDC problems, respectively.

**Prediction results of nonlinear forcing-type PDEs:** Figures 8 and 9 show the prediction results of Newton-PINet on some test cases for the convection-diffusion-reaction (CDR) problem and other nonlinear forcing-type PDEs.

**C.4. Comparison with baseline meta-learning PINNs on nonlinear benchmark problems**

We compare Newton-PINet with the baseline models from the recent study (Boudec et al., 2024) on a 1D nonlinear reaction-diffusion problem. The prediction results of Newton-PINet are shown in Fig. 10, and the detailed comparisons are

*Table 5.* Summary of model performance on nonlinear PDEs in this study. "NaN" indicates numerical overflow during computation.

| Problem | Meta-learning epoch | Meta-learning time cost | Inference time per test task | Test MSE of models | | |
|---|---|---|---|---|---|---|
| | | | | DNN | PINet | Newton-PINet |
| Burgers' | 1000 | 286s | 0.035s | 1.06e-4 | 1.10e-4 | 1.79e-6 |
| Generalized KdV | 1000 | 453s | 0.359s | 3.74e-2 | 1.49e-2 | 1.94e-3 |
| Kuramoto-Sivashinsky | 3000 | 1014s | 0.552s | 5.03e-1 | 106 | 3.45e-2 |
| Convection-diffusion-reaction | 1000 | 203s | 0.099s | 5.27e-1 | 4.65e-3 | 4.32e-5 |
| Klein-Gordon | 1000 | 105s | 0.065s | 1.04 | 1.88e-7 | 2.36e-7 |
| Hyperbolic heat | 2000 | 393s | 0.069s | 1.06e-1 | NaN | 1.50e-6 |
| Logarithmic heat | 2000 | 395s | 0.067s | 2.99e-1 | NaN | 2.16e-6 |
| Helmholtz | 2000 | 344s | 0.067s | 5.98 | NaN | 1.33e-3 |
| Parametric diffusion-reaction | 2000 | 342s | 0.047s | 3.43e-1 | 1.18e-8 | 1.60e-10 |
| Lid-driven cavity | 1000 | 290s | 0.40s | 1.42e-1 | 1.92e-4 | 1.67e-4 |

reported in Table 3 of the main text.

We further compare the performance of Newton-PINet with state-of-the-art gradient-based meta-learning PINNs (Penwarden et al., 2023) and Baldwinian-PINNs (Wong et al., 2025). For a fair comparison, we train an unsupervised Newton-PINet with a meta-learning loss of LSE and a hybrid Newton-PINet with a meta-learning loss of MSE, considering a single time-block setting. Table 6 summarizes the generalization performance and computational cost of Newton-PINet relative to these meta-learned PINN models. Compared to unsupervised gradient-based meta-PINNs, the unsupervised Newton-PINet consistently achieves lower test errors and shorter inference times. For instance, on the 2D parametric diffusion-reaction problem, Newton-PINet exhibits approximately a 179× improvement in generalization accuracy while requiring 7000× less computational time. Against gradient-free Baldwinian-PINN, which employs Picard to handle PDE nonlinear terms, Newton-PINet which employs the Newton linearization method, reduces test MSE by 2∼5 orders of magnitude across the benchmark problems and shortens single-task fine-tuning time by roughly one order of magnitude.

We also note that, in some cases, the unsupervised Newton-PINet achieves slightly higher accuracy than the hybrid Newton-PINet. This occurs for relatively simple nonlinear PDEs, where using an LSE-based meta-learning loss facilitates better convergence and generalization than MSE. Conversely, for more complex nonlinear PDEs, such as the KdV and K-S equations, adopting an MSE-based meta-learning loss leads to much better performance.

## D. Ablation studies

### D.1. Robustness of the meta-learned Tikhonov regularization parameter

In our method, the Tikhonov regularization parameter ($\lambda_{\text{reg}}$) is not manually tuned but meta-learned, enabling the model to identify a stable and task-agnostic regularization level across all training tasks. Once learned, $\lambda_{\text{reg}}$ remains fixed during test-time adaptation.

To evaluate its stability, using the Burgers' problem as an example, we additionally perform per-task optimization by initializing $\lambda_{\text{reg}}$ from the meta-learned value and further updating it for each test task with 300 epochs of gradient-based updates. As shown in Table 7, the meta-learned $\lambda_{\text{reg}}$ remains close to the per-task optimized values across a wide range of test tasks, and their generalization performance differs only marginally. This indicates that the meta-learned hyperparameter exhibits strong stability and low sensitivity during test-time adaptation.

### D.2. Effect of initial guess in nonlinear iterations

We conduct empirical ablations to evaluate how the initial field guess ($u_{\text{guess}}$) in nonlinear iterations influences model accuracy. Let's quantify the distance between $u_{\text{guess}}$ and the ground truth solution ($u_{\text{label}}$) using the $\ell_2$ norm ($\|u_{\text{guess}} -$

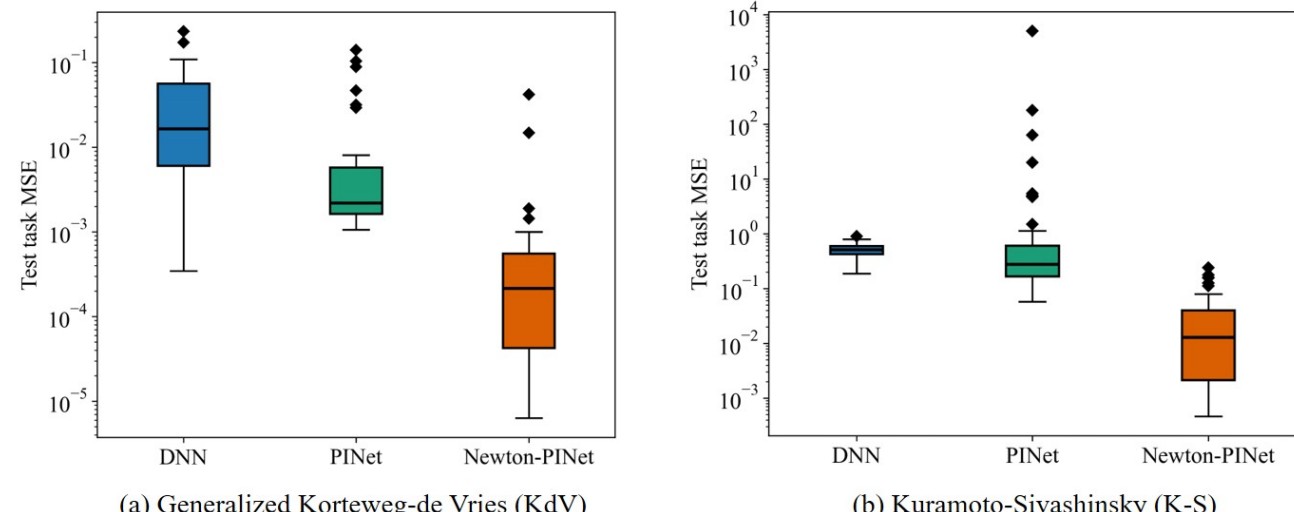

(a) Generalized Korteweg-de Vries (KdV)  (b) Kuramoto-Sivashinsky (K-S)

*Figure 4.* MSE distributions across all test tasks for different models on the generalized KdV and K-S problems.

$u_{\text{label}}\|_2$). To assess sensitivity to initialization, we test several representative choices for $u_{\text{guess}}$: setting $u_{\text{guess}}$ at all time steps equal to the initial field ($u_{t=0}$), all zeros field, all ones field, and normal random field.

As shown in Table 8, for the time-dependent Burgers' problem, using the $u_{\text{label}}$ as $u_{\text{guess}}$ leads to the lowest error, representing the convergence limit of our method. The $u_{t=0}$ strategy keeps $u_{\text{guess}}$ close to the $u_{\text{label}}$ (lowest non-zero distance) and achieves the best accuracy among the evaluated initialization choices. As shown in the Fig. 11, the converged error starting from $u_{t=0}$ is comparable to that obtained when using $u_{\text{label}}$ as the initial guess. For the more chaotic K-S system, attempting to solve a single time-block with the initialization of $u_0$ proved very challenging. This is because a simple initial condition can evolve into highly complex structures over time, causing this $u_{\text{guess}}$ to deviate from $u_{\text{label}}$. To address this, we employed a 15-block temporal decomposition strategy. The block-wise inference generally makes the $u_{\text{guess}}$ for each sub-block closer to the $u_{\text{label}}$, thereby improving the convergence of the least-squares solve. As Table 8 shows, the test errors under different initial guesses are comparable. This indicates that the temporal block decomposition mitigates the sensitivity of the least-squares solution to the choice of $u_{\text{guess}}$. We further tested the time-independent lid-driven cavity (LDC) problem and found that zero initialization achieves the lowest test error.

Our experiments show that for the strongly nonlinear/chaotic K-S equations, even a relatively small initialization distance (10–20) can lead to significantly worse convergence, whereas tasks like the Burgers' equation and LDC problems are much less sensitive. Crucially, our empirical observations unveil a few strategies to mitigate this sensitivity to initialization. Setting $u_{\text{guess}}$ at all time steps equal to the initial field ($u_{t=0}$) can help provide high accuracy for time-dependent problems, while a temporal block decomposition strategy may be employed for increasingly more complex nonlinear time-dependent systems like the K-S problem.

### D.3. Effect of time-blocking strategies

We evaluate the effect of different numbers of time blocks on the average test error taking the convection-diffusion-reaction (CDR) problem as an example. As shown in Table 9, without temporal decomposition (i.e., using a single time block), the model achieves fast inference but suffers from large errors. This is because the initial guess in the nonlinear iteration may deviate significantly from the true solution, causing poor least-squares accuracy. To address this issue, we apply a temporal domain decomposition strategy: starting from the known initial condition, the final prediction of each block serves as the initial guess for the next. This ensures that each block starts more closely to the true dynamics, enabling reliable long-horizon prediction. As shown in the table, increasing the number of blocks can substantially reduce error, though at the cost of longer inference time. For this problem, a 4-block configuration is adopted to balance the accuracy and efficiency.

*Table 6.* Model comparison on nonlinear benchmark problems. Baselines are reported from (Penwarden et al., 2023; Wong et al., 2025). "–" denotes results not reported in the references.

| Meta-learning PINNs | | | Nonlinear benchmark problems (number of test tasks) | | | |
|---|---|---|---|---|---|---|
| | | | Heat (64) | Allen-Cahn (32) | Diffusion -reaction (64) | 6D diffusion -reaction (100) |
| Unsupervised | Random | Error[1] | 0.0052 | 0.015 | 0.011 | 0.0022 |
| | | Time[2] | 156 | 496 | 1073 | 612 |
| | MAML | Error[1] | 0.0045 | – | – | – |
| | | Time[2] | 188 | – | – | – |
| | Center | Error[1] | 0.0045 | 0.012 | 0.0095 | 0.0018 |
| | | Time[2] | 96 | 201 | 426 | 494 |
| | Multitask | Error[1] | 0.0048 | 0.012 | 0.009 | 0.0017 |
| | | Time[2] | 49 | 120 | 243 | 431 |
| | LMC | Error[1] | 0.0049 | 0.011 | 0.0091 | 0.0015 |
| | | Time[2] | 59 | 120 | 302 | 428 |
| | RBF | Error[1] | 0.0044 | 0.012 | 0.0081 | 0.0017 |
| | | Time[2] | 35 | 68 | 280 | 375 |
| | Polynomial | Error[1] | 0.0046 | 0.012 | 0.0085 | 0.0018 |
| | | Time[2] | 38 | 44 | 249 | 496 |
| | **Newton-PINet** (Ours) | Rel. $L^2$ | 4.0e-6 | 4.2e-7 | 1.4e-4 | 9.5e-6 |
| | | Time[3] | 0.06 | 0.044 | 0.047 | 0.049 |
| Hybrid | Baldwinian-PINN | Rel. $L^2$ | 6.0e-4 | 9.5e-4 | 1.3e-4 | 1.9e-4 |
| | | MSE | 1.3e-7 | 2.0e-7 | 1.2e-8 | 1.6e-8 |
| | | Time[3] | 0.87 | 1.19 | 0.58 | 0.6 |
| | **Newton-PINet** (Ours) | Rel. $L^2$ | 7.3e-6 | 3.7e-6 | 1.4e-4 | 1.3e-5 |
| | | MSE | 1.7e-11 | 2.5e-12 | 6.8e-10 | 1.6e-10 |
| | | Time[3] | 0.062 | 0.044 | 0.047 | 0.047 |

[1] Relative $L^2$ errors obtained after 500 iterations of fine-tuning using ADAM followed by L-BFGS, as described in (Penwarden et al., 2023).
[2] Only the L-BFGS optimization time (s) is reported in (Penwarden et al., 2023).
[3] Mean inference time per test task (s), on a Tesla V100 GPU.

### D.4. Effect of initial condition weight on drift at temporal block boundaries

As can be seen in Fig. 3, the CDR prediction results exhibit noticeable drift at temporal block boundaries. Upon investigation, the drift at boundaries is actually due to an under-weighted enforcement of the initial condition (IC) relative to the PDE and boundary condition (BC) in Tikhonov regularization during the meta-learning and test time. After the ablation study, as shown in Table 10, we found that increasing the IC weight ($\lambda_{ic}$) from 1 to 2 can reduce the test error. Figure 12 illustrates that this IC weight adjustment can effectively mitigate the drift at block boundaries.

### D.5. Model performance under other types of boundary conditions

As shown in Table 11, we test the model performance for Dirichlet, Neumann, and mixed BCs on the hyperbolic heat and the Klein-Gordon problems. As shown in Table 12, we also evaluate the model for discontinuous BCs on the 2D lid-driven cavity Navier-Stokes problem, where the velocity field exhibits corner discontinuities at the moving lid. These experiments demonstrate that the proposed framework is both theoretically and practically applicable to a broad range of BC types, beyond periodic and Dirichlet settings.

*Table 7.* Sensitivity of the meta-learned $\lambda_{\mathrm{reg}}$ to task adaptation in test inference. For meta-learning, $\lambda_{\mathrm{reg}}$ is learned during meta-training and kept fixed during testing. For Per-task optimization, $\lambda_{\mathrm{reg}}$ is initialized from the meta-learned model parameters and then optimized for each test task through gradient-based updates.

| Method | | $\gamma$ (test tasks of Burgers' problem) | | | | | |
| --- | --- | --- | --- | --- | --- | --- | --- |
| | | 0.001 | 0.003 | 0.004 | 0.006 | ... | 0.047 |
| Meta-learning | $\lambda_{\mathrm{reg}}$ | | | 7.57e-4 | | | |
| | MSE | 3.30e-5 | 2.07e-6 | 1.44e-6 | 4.59e-7 | ... | 7.54e-8 |
| Per-task optimization | $\lambda_{\mathrm{reg}}$ | 2.07e-5 | 4.34e-4 | 4.81e-4 | 3.26e-4 | ... | 9.79e-5 |
| | MSE | 3.21e-5 | 2.01e-6 | 1.44e-6 | 4.69e-7 | ... | 7.07e-8 |

*Table 8.* Effect of different initial guesses for the nonlinear iterations on test errors.

| Problem | Metrics | Initial guess for the nonlinear iteration ($u_{\mathrm{guess}}$) | | | | |
| --- | --- | --- | --- | --- | --- | --- |
| | | Ground truth | All $u_{t=0}$ (ours) | All zeros | All ones | Normal random |
| Burgers' | Distance | 0 | 26.4 | 45.9 | 93.2 | 93.6 |
| | MSE | 1.24e-6 | 1.24e-6 | 1.24e-6 | 1.11e-04 | 4.84e-3 |
| | Relative $L^2$ | 1.04e-3 | 1.04e-3 | 1.04e-3 | 9.61e-03 | 4.96e-2 |
| K-S | Distance | 0 | 3.3 | 13.3 | 23.9 | 24.4 |
| | MSE | 2.84e-2 | 2.83e-2 | 2.91e-2 | 2.91e-2 | 3.08e-2 |
| | Relative $L^2$ | 1.76e-1 | 1.76e-1 | 1.79e-1 | 1.79e-1 | 1.86e-1 |
| LDC | Distance | 0 | – | 10.5 | 51.7 | 51.3 |
| | MSE | 1.58e-4 | – | 1.67e-4 | 1.76e-3 | 1.86e-4 |
| | Relative $L^2$ | 5.54e-2 | – | 5.61e-2 | 1.32e-1 | 5.69e-2 |

## D.6. Ablation studies on model architecture

Newton-PINet employs a skip-connected multi-layer architecture with Tikhonov regularization, Sinusoidal feature embeddings at the input, and $\sin(\cdot)$ activations in all hidden layers. In this section, we perform ablation studies to examine the impact of these design choices and other hyperparameters. All test errors in the following tables are aggregated from 5 individual runs.

**Effect of network depth:** We first study the impact of network depth using the Burgers' and CDR problems, with and without skip connections (Table 13). For comparison, Sinusoidal feature embeddings are always included at the input, and all hidden layers adopt $\sin(\cdot)$ activations. The number of neurons per layer is set to 512, 256, 170, and 128 for depths of 1∼4, respectively, so that the last-layer width remains around 512 when skip connections are included. The results show that shallow networks (1 layer) fail to achieve satisfactory accuracy in both problems. Increasing depth without skip connections often deteriorates generalization. In contrast, with skip connections, the model consistently maintains high accuracy across all depths, confirming the stabilizing effect of skip connections and their robustness to depth variations. This property greatly simplifies hyperparameter tuning compared to standard PINNs.

**Effect of network width:** Next, we fix the depth to 4 layers and compare the effect of different per-layer widths, with and without skip connections (Table 14). Across both Burgers' and CDR problems, incorporating skip connections consistently improves accuracy under almost all width settings. An exception occurs for the case where all hidden layers have 256 neurons, where skip connections slightly degrade performance—likely due to the substantially increased final-layer width, which makes the Tikhonov regularization computation numerically more challenging. Overall, skip connections improve robustness to width variations.

**Effect of mesh resolution:** We then examine performance under different mesh resolutions, with consistent resolutions applied during meta-training and testing (Table 15). With skip connections, the model exhibits stronger adaptability to

*Table 9.* Ablation study for different time-blocking strategies.

| Metrics | Number of blocks | | | | | | |
|---|---|---|---|---|---|---|---|
| | 1 | 2 | **4** | 6 | 10 | 15 | 20 |
| Test MSE | 3.56e-3 | 4.46e-3 | **9.57e-6** | 2.93e-6 | 9.07e-7 | 1.05e-6 | 5.78e-7 |
| Test relative $L^2$ | 2.23e-1 | 3.96e-2 | **3.56e-3** | 1.80e-3 | 1.13e-3 | 1.12e-3 | 1.08e-3 |
| Inference time per task (s) | 3.74e-2 | 6.82e-2 | **1.01e-1** | 4.04e-1 | 6.77e-1 | 1.26 | 1.66 |

*Table 10.* Effect of initial condition (IC) weight in the Tikhonov regularization on test errors.

| Errors | Weight parameter for initial condition ($\lambda_{\text{ic}}$) | | | | |
|---|---|---|---|---|---|
| | 1 | 1.5 | 2 | 5 | 10 |
| Test MSE | 9.57e-6 | 6.56e-6 | 5.75e-6 | 5.77e-6 | 6.46e-6 |
| Test relative $L^2$ | 3.56e-3 | 2.89e-3 | 2.67e-3 | 2.67e-3 | 2.80e-3 |

resolution changes, particularly in the CDR problem. Notably, different resolutions alter the size of the matrices, which can influence numerical stability. Nevertheless, skip connections help preserve accuracy under these settings.

**Effect of activation functions and Sinusoidal embeddings:** We further compare activation functions under varying depths, with skip connections included (Table 16). For $\tanh(\cdot)$ and Gaussian activations, Sinusoidal embeddings are not used, while for $\sin(\cdot)$ activations they are applied at the input. The combination of Sinusoidal embeddings with $\sin(\cdot)$ consistently outperforms other activations across different depths, highlighting the effectiveness of this design choice for nonlinear PDE learning.

**Effect of weight initialization and learning rate:** During meta-learning, we update the hidden-layer weights ($\tilde{w}$) and other learning parameters via gradient descent. Here, we investigate the impact of weight initialization schemes and learning rate ($\eta$) settings on the model generalization (Table 17). Results indicate that He initialization together with a smaller initial learning rate leads to the most stable and accurate generalization performance, whereas larger learning rates often cause unstable training or accuracy degradation.

**In summary:** Newton-PINet benefits significantly from skip connections, which enhance robustness to depth, width, and mesh resolution, thereby reducing the need for extensive hyperparameter tuning. Sinusoidal feature embeddings with $\sin(\cdot)$ activations further boost accuracy across depths, and careful initialization and learning rate selection improve stability. Collectively, these results explain why Newton-PINet maintains reliable generalization across diverse nonlinear PDE problems.

# E. Practical scaling discussion of Newton-PINet

**Effectiveness of Tikhonov regularization:** The scalability of the Tikhonov solution is determined by the matrix $\mathbf{A}$. Its rows grow with the total number of PDE, IC, and BC residual samples. Its column size is determined by the product of the pre-final feature dimension and the output dimension ($D_o$). The pre-final feature dimension equals the total number of neurons in the $L-1$ hidden layers (each of width $N_n$) due to the skip connections. This results in $\mathbf{A} \in \mathbb{R}^{(n_{\text{pde}}+n_{\text{ic}}+n_{\text{bc}}) \times D_o N_n (L-1)}$.

For 2D/3D or multi-physics systems (e.g., Navier-Stokes), the increase in the number of PDE equations requires more residual sampling points, while a higher output dimensionality enlarges the number of columns, making $\mathbf{A}$ larger and more challenging to solve, particularly during Newton iterations for nonlinear problems.

Our strategy is to moderately control the pre-final feature dimension, for example, by decreasing the number of neurons in each layer, thereby reducing the number of columns of $\mathbf{A}$ and ensuring that the least-squares problem remains reasonably overdetermined. Since the Tikhonov update $(\lambda_{\text{reg}}\mathbf{I} + \mathbf{A}^T\mathbf{A})\boldsymbol{w} = \mathbf{A}^T\mathbf{b}$ involves only $\mathbf{A}^T\mathbf{A} \in \mathbb{R}^{D_o N_n (L-1) \times D_o N_n (L-1)}$,

*Table 11.* Model performance under different boundary conditions.

| Problem | Dirichlet | | | Neumann | | | Mixed | | |
|---|---|---|---|---|---|---|---|---|---|
| | Time[1] | MSE | Rel. $L^2$ | Time[1] | MSE | Rel. $L^2$ | Time[1] | MSE | Rel. $L^2$ |
| Heat | 0.067 | 1.69e-6 | 1.76e-3 | 0.071 | 1.80e-5 | 5.84e-3 | 0.086 | 5.35e-5 | 1.04e-2 |
| Klein-Gordon | 0.064 | 2.36e-7 | 3.86e-4 | 0.084 | 2.34e-6 | 1.16e-3 | 0.082 | 4.05e-6 | 1.13e-3 |

1. Inference time per test task (s)

*Table 12.* Model performance for 2D lid-driven cavity (LDC) under discontinuous boundary conditions.

| Problem | Inference time per test task (s) | Test MSE | Test Relative $L^2$ |
|---|---|---|---|
| 2D LDC | 0.40 | 2.30e-4 | 6.04e-2 |

its computational cost depends primarily on the feature dimension, as opposed to the potentially very large number of residual samples in high-dimensional or multiphysics domains. This keeps the Tikhonov solver tractable as long as the feature dimension is properly controlled. In our experiments, the Newton-PINet achieves high accuracy without requiring a large pre-final feature dimension (i.e., no. features $\ll$ no. collocation points).

**Time-blocking strategy:** In addition to the matrix size, the scalability of our model for nonlinear problems also depends on the quality of the initial guess for the iterative updates. Similar to the theoretical behavior of the Newton method, if the initial guess deviates too far from the true solution, more Newton iterations are required to achieve reasonable convergence. This challenge becomes particularly severe for time-dependent chaotic systems, such as the Kuramoto-Sivashinsky (K-S) equation, where even simple initial conditions can quickly evolve into highly complex spatiotemporal patterns. Consequently, it is difficult to obtain an initial guess close to the solution. We can adopt the temporal domain decomposition strategy: starting from the known initial condition, the final prediction of a previous block is used as the initial guess for the next block. This ensures that each block starts more closely to the true dynamics, improving the stability and accuracy of the least-squares updates without using large Newton iterations, and enabling reliable long-horizon prediction.

**Low-rank strategy:** We would like to clarify that although the task adaptation step is formulated as a least-squares problem, the matrix $\mathbf{A}$ in our method is not sparse in the traditional numerical PDE solver sense. Because the features are generated by a multilayer neural representation with skip connections, each row of $\mathbf{A}$ contains dense network features rather than the typical discretization-induced sparsity and structure. As a consequence, the rank and conditioning of our least-squares system is determined jointly by (i) the density and distribution of collocation points, (ii) the Tikhonov parameter, and more importantly (iii) the quantity and expressiveness of hidden-layer weights that shape the feature space. **In this instance, the practical scaling for large collocation sets may derive more from the fact that one can still control the cost of the solve through meta-learning a better (yet minimal) set of pre-final layer features.**

For this reason, low-rank or iterative solvers designed for large sparse PDE matrices (e.g., LU decomposition or Krylov solvers) may not offer clear advantages in our setting. We experimented with SVD-based solvers, LU, and Cholesky factorizations; none achieved better accuracy-stability trade-offs than the Tikhonov formulation used in our method.

Nonetheless, a more comprehensive investigation of scalable iterative schemes tailored to dense neural-feature matrices is an interesting direction for future work.

# F. Use of Large Language Models (LLMs)

In preparing this paper, we used large language models (LLMs) solely as an auxiliary tool for writing support. Specifically, the LLM assisted in refining some English expressions and improving the formatting of tables. The research ideas, literature review, methodological development, code implementation, figures, and experimental results were entirely conceived and executed by the authors without the involvement of LLMs.

*Table 13.* Test relative $L^2$ errors of model with (w/) and without (w/o) skip connections under different network depths.

| Problem | Skip connections | Network depth | | | |
|---|---|---|---|---|---|
| | | 1 layer | 2 layers | 3 layers | 4 layers |
| Burgers' | w/o | 1.04e-2 | 3.75e-4 | 4.77e-5 | 5.02e-5 |
| | | ±1.22e-2 | ±1.50e-3 | ±1.20e-4 | ±1.05e-4 |
| | w/ | 1.04e-2 | 1.43e-5 | 1.93e-6 | 1.79e-6 |
| | | ±1.22e-2 | ±7.21e-5 | ±9.24e-6 | ±7.98e-6 |
| CDR | w/o | 4.70e-3 | 7.51e-5 | 6.80e-4 | 2.12e-3 |
| | | ±1.62e-3 | ±7.46e-5 | ±7.30e-4 | ±1.97e-3 |
| | w/ | 4.70e-3 | 5.43e-5 | 5.64e-5 | 4.32e-5 |
| | | ±1.62e-3 | ±4.58e-5 | ±5.88e-5 | ±4.18e-5 |

*Table 14.* Test relative $L^2$ errors with (w/) and without (w/o) skip connections under different network widths (from top to bottom, corresponding to layer 1∼4).

| Problem | Skip connections | Number of nodes in each layer | | | | | | |
|---|---|---|---|---|---|---|---|---|
| | | 256 256 256 256 | 128 128 128 128 | 64 64 64 64 | 32 32 32 32 | 16 16 16 16 | 16 32 64 128 | 128 64 32 16 |
| Burgers' | w/o | 9.13e-6 | 5.02e-5 | 1.01e-3 | 1.24e-3 | 1.77e-3 | 4.87e-4 | 1.14e-3 |
| | | ±1.87e-5 | ±1.05e-4 | ±2.06e-3 | ±2.12e-3 | ±2.38e-3 | ±8.03e-4 | ±1.95e-3 |
| | w/ | 2.19e-6 | 1.79e-6 | 4.12e-6 | 8.14e-5 | 2.03e-3 | 3.17e-4 | 5.41e-6 |
| | | ±7.43e-6 | ±7.98e-6 | ±1.44e-5 | ±1.11e-4 | ±2.78e-3 | ±5.66e-4 | ±1.66e-5 |
| CDR | w/o | 3.88e-4 | 2.12e-3 | 4.80e-3 | 5.98e-2 | 1.05e-1 | 1.34e-3 | 1.24e-1 |
| | | ±3.44e-4 | ±1.97e-3 | ±3.79e-3 | ±8.14e-2 | ±6.29e-2 | ±8.42e-4 | ±7.24e-2 |
| | w/ | 1.10e-3 | 4.32e-5 | 2.25e-4 | 7.73e-4 | 6.48e-3 | 2.02e-4 | 4.55e-4 |
| | | ±2.20e-3 | ±4.18e-5 | ±2.14e-4 | ±4.77e-4 | ±8.32e-3 | ±1.82e-4 | ±3.47e-4 |

*Table 15.* Test relative $L^2$ errors with (w/) and without (w/o) skip connections under different mesh resolutions

| Problem | Skip connections | Mesh resolution $(t, x)$ | | | | |
|---|---|---|---|---|---|---|
| | | 11×29 | 13×33 | 17×43 | 26×65 | 51×129 |
| Burgers' | w/o | 4.17e-5 | 7.19e-6 | 2.23e-5 | 1.27e-5 | 5.02e-5 |
| | | ±4.57e-5 | ±1.52e-5 | ±2.30e-5 | ±1.74e-5 | ±1.05e-4 |
| | w/ | 2.57e-4 | 5.54e-6 | 2.80e-5 | 1.49e-6 | 1.79e-6 |
| | | ±7.24e-4 | ±1.62e-5 | ±4.61e-6 | ±7.00e-6 | ±7.98e-6 |
| | | 11×29 | 21×49 | 41×69 | 71×99 | 101×129 |
| CDR | w/o | 6.34e-2 | 3.08e-3 | 2.26e-3 | 1.51e-3 | 2.12e-3 |
| | | ±9.09e-2 | ±2.42e-3 | ±1.78e-3 | ±1.16e-3 | ±1.97e-3 |
| | w/ | 3.72e-2 | 3.27e-3 | 4.03e-5 | 3.36e-5 | 4.32e-5 |
| | | ±3.85e-2 | ±4.55e-3 | ±1.44e-5 | ±3.13e-5 | ±4.18e-5 |

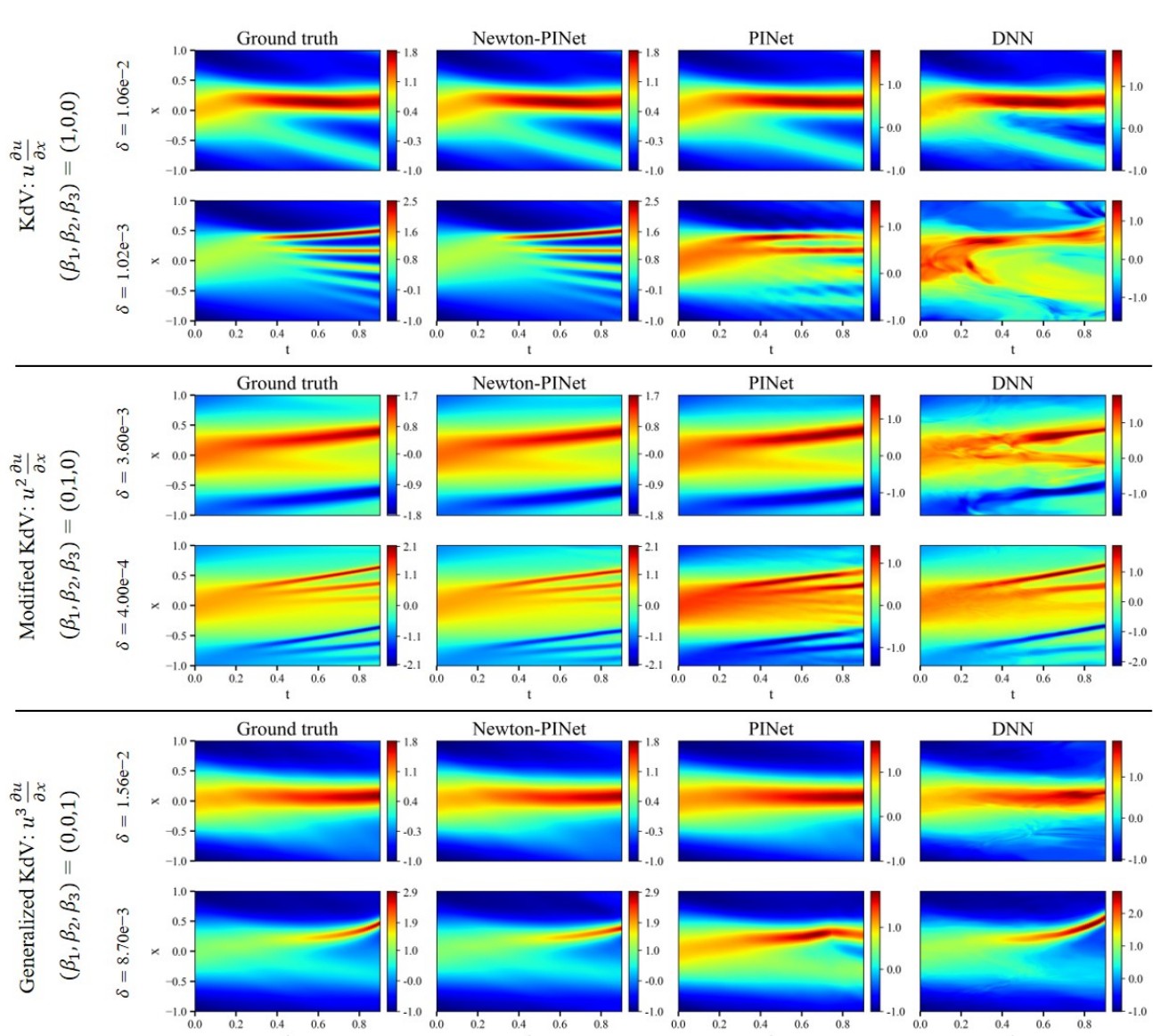

*Figure 5.* Prediction results of Newton-PINet for the generalized KdV problem.

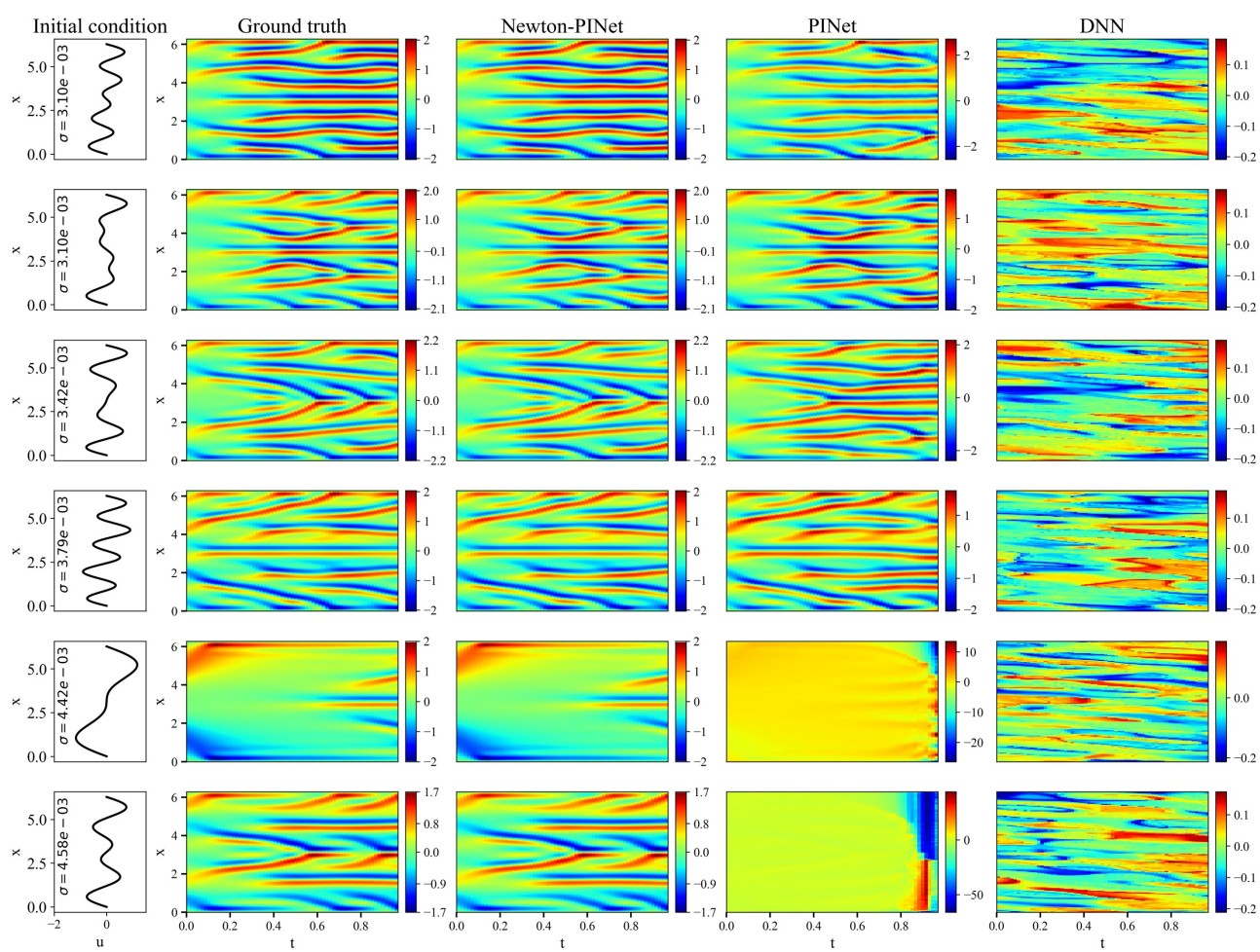

*Figure 6.* Prediction results of Newton-PINet for the K-S problem.

*Table 16.* Test relative $L^2$ errors of models with different activation functions under varying network depths

| Problem | Activation function | Network depth | | | |
|---------|---------------------|---------|----------|----------|----------|
| | | 1 layer | 2 layers | 3 layers | 4 layers |
| Burgers' | tanh | 1.07e-2 | 1.30e-4 | 6.39e-4 | 7.13e-5 |
| | | ±1.30e-2 | ±2.42e-4 | ±2.54e-3 | ±1.60e-4 |
| | Gaussian | 1.14e-2 | 4.64e-3 | 8.27e-6 | 3.28e-6 |
| | | ±1.31e-2 | ±8.95e-3 | ±3.06e-5 | ±1.26e-5 |
| | sin | 1.04e-2 | 1.43e-5 | 1.93e-6 | 1.79e-6 |
| | | ±1.22e-2 | ±7.21e-5 | ±9.24e-6 | ±7.98e-6 |
| CDR | tanh | 3.94e-2 | 1.84e-3 | 6.09e-3 | 4.16e-2 |
| | | ±1.33e-2 | ±2.18e-3 | ±1.16e-2 | ±8.27e-2 |
| | Gaussian | 3.60e-2 | 5.62e-3 | 6.62e-4 | 2.57e-4 |
| | | ±2.11e-2 | ±5.25e-3 | ±7.16e-4 | ±2.01e-4 |
| | sin | 4.70e-3 | 5.43e-5 | 5.64e-5 | 4.32e-5 |
| | | ±1.62e-3 | ±4.58e-5 | ±5.88e-5 | ±4.18e-5 |

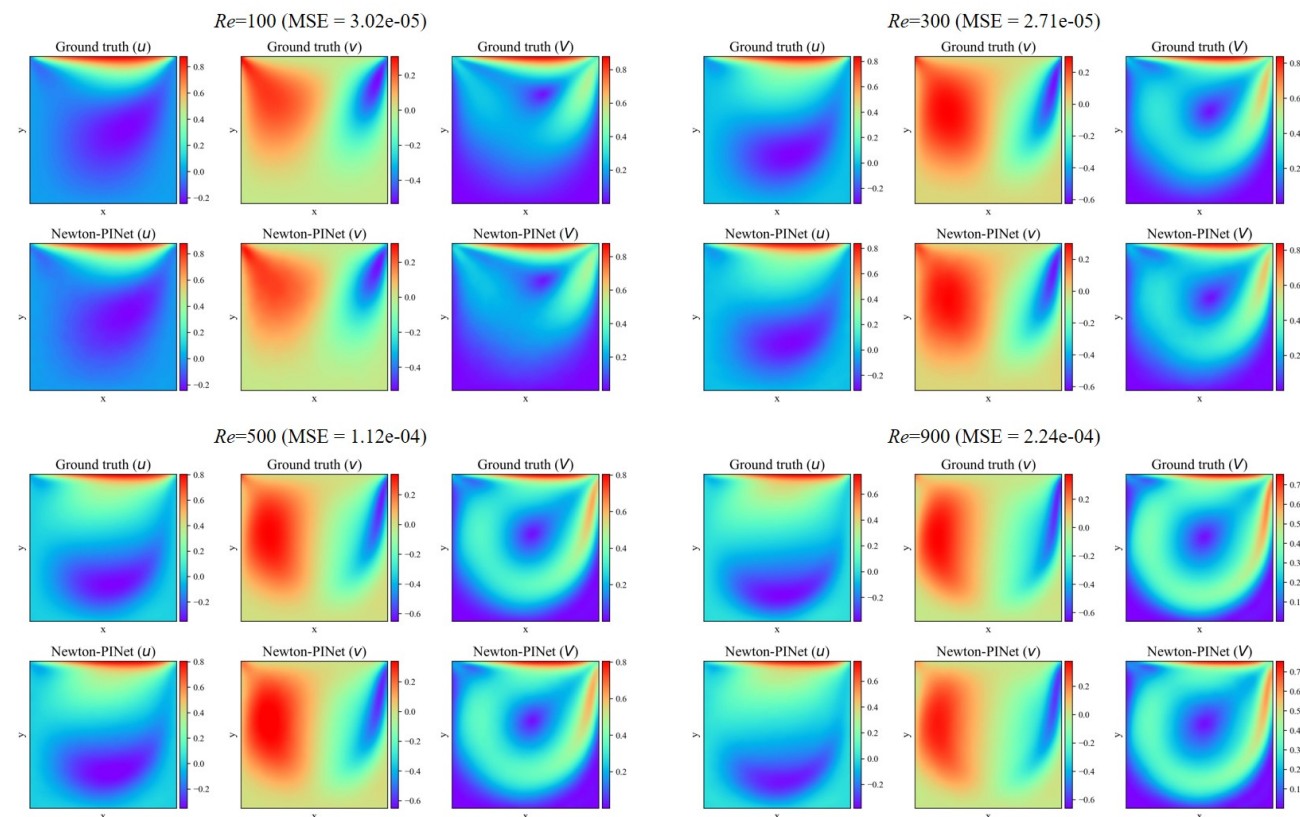

*Figure 7.* Prediction result of Newton-PINet for the 2D lid-driven cavity problem. Velocity magnitude is computed as $V = \sqrt{u^2 + v^2}$.

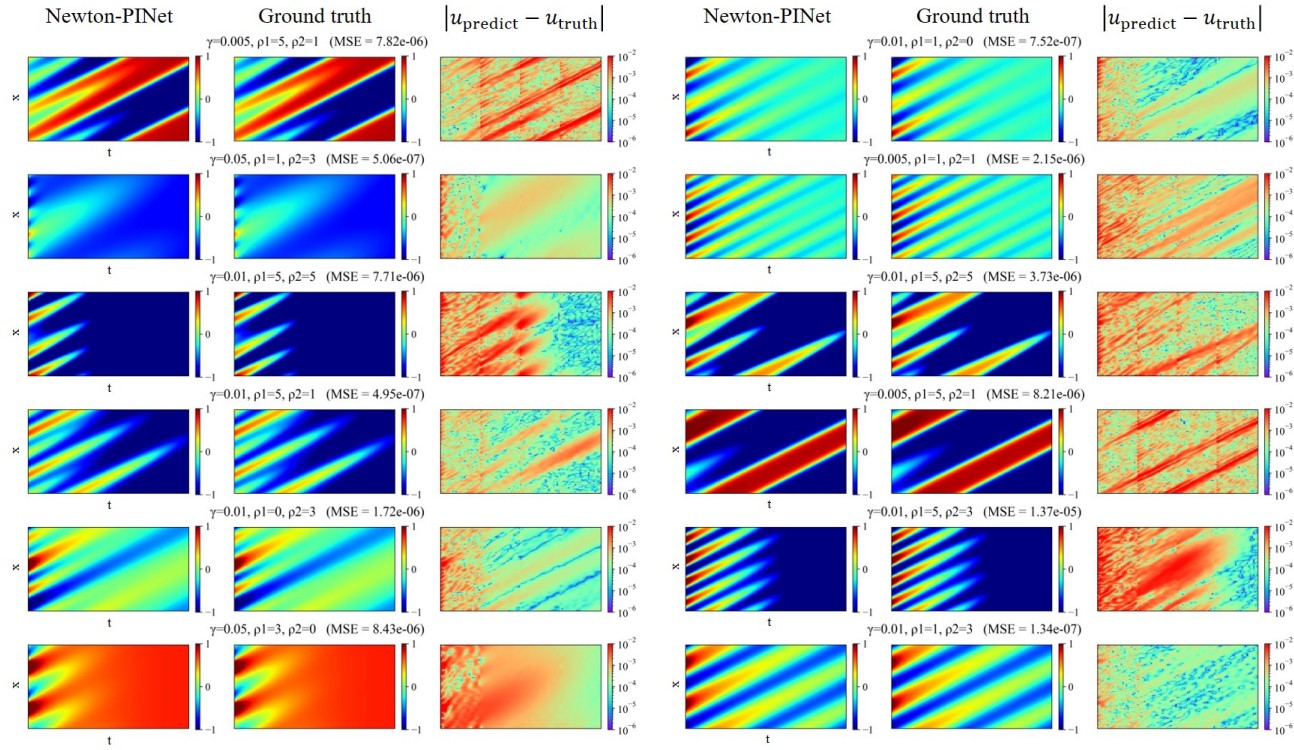

*Figure 8.* Prediction results of Newton-PINet for the CDR problem.

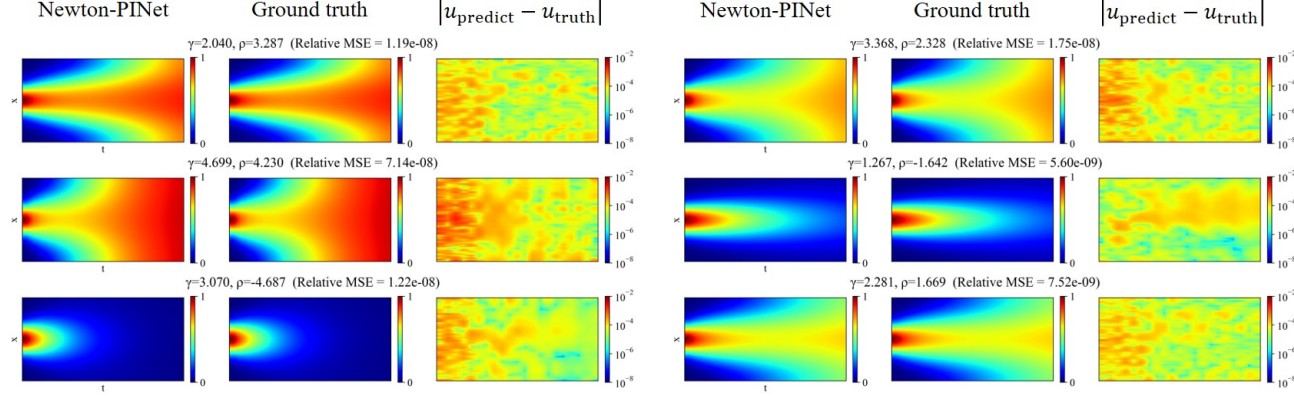

*Figure 9.* Prediction results of Newton-PINet for the other 5 nonlinear PDE problems.

*Figure 10.* Prediction results of Newton-PINet for the 1D nonlinear reaction-diffusion benchmark problem.

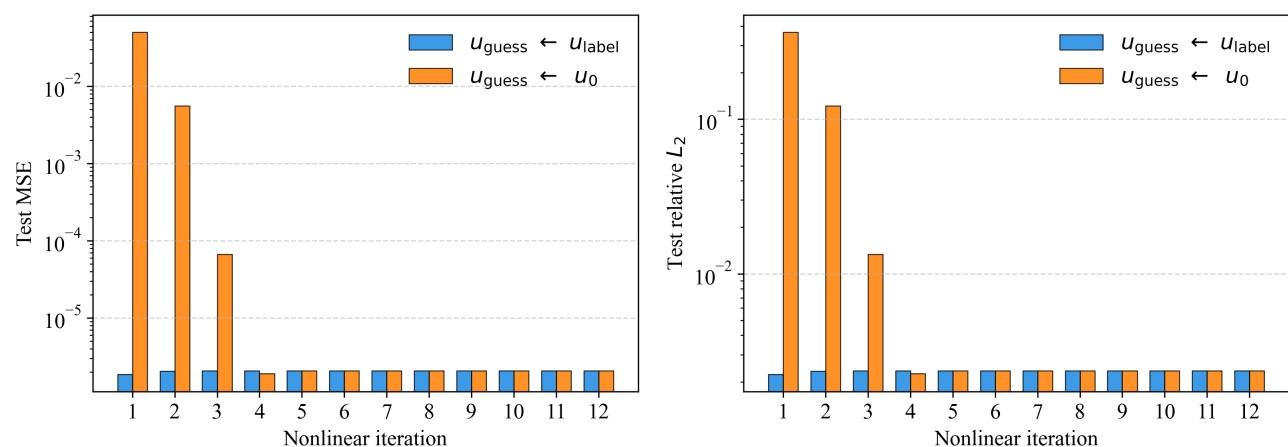

*Figure 11.* Convergence comparison of single-task generalization starting from the true field versus $u_0$ as initial guesses.

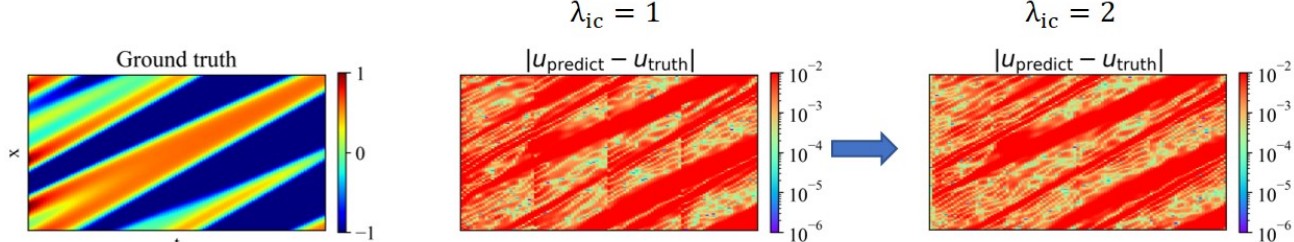

*Figure 12.* Mitigation of drift at block boundaries via adjustment of the initial condition (IC) weight in the Tikhonov regularization (4-time-block for CDR inference).

*Table 17.* Test relative $L^2$ errors under different weight initialization methods and initial learning rates

| Problem | Weight initialization | Initial learning rate | | |
|---|---|---|---|---|
| | | 0.0005 | 0.005 | 0.05 |
| Burgers' | Xavier | 7.57e-5 ±1.62e-4 | 2.83e-6 ±9.33e-6 | 2.09e-3 ±5.54e-3 |
| | He | 1.57e-5 ±7.89e-5 | 1.79e-6 ±7.98e-6 | 2.69e-3 ±4.93e-3 |
| CDR | Xavier | 2.62e-2 ±2.17e-2 | 5.79e-4 ±4.69e-4 | 1.11e-3 ±2.20e-3 |
| | He | 2.78e-4 ±3.27e-4 | 4.32e-5 ±4.18e-5 | 1.68e-2 ±3.20e-2 |

