# OpenReview forum: "Newton-PINet: A fast physics-informed neural network with Newton linearization for meta-learning nonlinear PDEs"
_ICML.cc/2026/Conference — Submitted to ICML 2026_

### Official Review · Reviewer_UYkA · 2026-02-27

**Soundness:** 2
**Presentation:** 1
**Significance:** 3
**Originality:** 3
**Overall Recommendation:** 4
**Confidence:** 3

**Summary:**

The paper proposes a meta-learning framework, called Newton-PINet, to learn a physics-informed neural operator that solves a family of PDEs. The method pretrain the weights of hidden layers once and adapts to new PDE instances by solving only the weight of output layer via Tikhonov regularization, in which the nonlinear dynamics is replaced by Newton linearization.
The experiments across diverse 1D/2D nonlinear PDEs benchmarks demonstrate significant improvements in data efficiency, training time, and adaptation time.

**Compliance With Llm Reviewing Policy:**

Affirmed.

**Final Justification:**

The rebuttal addresses my major concerns regarding grid dependency, code availability, and the mathematical derivation of the core regularized linear system. While the scientific contribution meets the acceptance bar for ICML, the presentation of the current submission remains difficult to follow. I strongly recommend that the authors incorporate the clarifications provided in the rebuttal into the final manuscript to improve readability and broaden the paper's accessibility.

**Key Questions For Authors:**

1. In equation (7), the PDE operator, IC, and BC are all applied on the pre-final output $f_j$. In this sense, do you view each $f_j$ is some “basis” of the solution space and aim to find the best linear combination through the least squares problem?

2. I’m still confused by equation (7) and Figure 1 regarding $A$ and $b$. In the Nonlinear iterations box of Figure 1, there is a connection from "$u = f(x, t; \tilde{w})^T w$" to "Least-squares matrix and vector $A$ and $b$". Why does new $u$ result in new $A$ and $b$? Equation (7) seems to suggest that $A$ and $b$ depend on pre-final output $f$ only, which does not seem to change after new weights.

3. How many collocation points are used, and what is the size of the resulting least-squares problems in these experiments? Does the method exhibit dependency on the grid or collocation points? If so, would this grid dependency subject the method to the curse of dimensionality, thereby undermining one of the primary advantages of neural operators—namely, their grid-free nature?

If the authors could carefully address my concerns and improves the presentation, I will be willing to increase my score.

**Limitations:**

yes

**Strengths And Weaknesses:**

### Strengths
- The experiments show significant improvements in terms of MSE, training time, and adaptation time across diverse 1D/2D nonlinear PDEs benchmarks.

- The combination of PINNs, Tikhonov regularization, and Newton linearization is novel for the current problem of learning physics-informed neural operators.

- Gradient-free adaption reduces the computational time for new PDE instances.

### Weaknesses

- The presentation of the main paper is not clear enough for readers to understand the technical details of the proposed method, especially the derivation of matrix $A$ and vector $b$ in the least-squares problem for the output layer weights. Though the appendix provides more details, the authors is responsible for making the main paper self-contained and clear to readers. For example, equation (7) in Appendix specifically defines the matrix $A$ and vector $b$ and thus should be included in the main paper for better clarity, though the current expression of (7) is lengthy and should be simplified for better readability in the main paper.

- The code of the paper is not available. Given the technicality of the proposed method and less clarity in the presentation, it is hard for readers to reproduce the results without the code.

- The least-squares problem for the output layer weights seems to scale with the number of collocation points, which may limit the scalability of the method to high-dimensional PDEs or large datasets. The paper does not discuss this potential limitation or provide any analysis on the computational complexity of the method.

---

> ### Author Rebuttal · Authors · 2026-03-31
>
> We sincerely thank the reviewer for recognizing the value of our work.
>
> **Response to W1:**
>
> We could incorporate the derivation of matrix $\mathbf{A}$ and vector $\mathbf{b}$ (based on a simplified version of Eq. 7) into the Methodology section in the revised manuscript.
>
> **Response to W2:**
>
> We would like to humbly clarify that the code and data were included in the Supplementary Material (zip file) during submission. If they remain inaccessible, we can provide an anonymous GitHub link during the discussion stage (if permitted) and update the manuscript with a GitHub link upon successful publication.
>
> **Response to W3:**
>
> The computational complexity and scalability of the model are discussed in detail in *Appendix E* (page 28–29), but were not elaborated in the main text due to space constraints. We will further expand this discussion and add a brief section in the main text. To save space for your other comments, we respectfully refer you to our responses to W1 and W2 of **Reviewer 3 [g64x]**, which address a similar concern.
>
> **Response to Q1:**
>
> We appreciate your thoughtful observation. We indeed view the features extracted by the neural network as a basis for the solution space. In fact, this is the primary motivation behind our adopted concatenated feature architecture (referred to as skip-connections). By concatenating features from all hidden layers before the output layer, we effectively expand and enrich this basis functional space.
>
> Much like incorporating higher-order terms in a polynomial or Chebyshev expansion, each additional hidden layer in our network introduces new nonlinear features. This mechanism significantly enhances the expressivity of the basis while maintaining a moderate number of nodes per layer, thereby reducing truncation effects and ensuring the stability and precision of the least-squares computation.
>
> **Response to Q2:**
>
> The nonlinear iteration applies only to nonlinear PDEs, and the change in the matrix $\mathbf{A}$ arises from the linearization using $u$ and its associated derivatives from the previous iteration. The reviewer's understanding is indeed correct for linear PDEs, where $\mathbf{A}$ and $\mathbf{b}$ remain unchanged and the linear system can be solved in a single step (due to space limitations, kindly refer to the description between Eqs. (9)–(11) on page 13 for details).
>
> **Response to Q3:**
>
> We have summarized the number of collocation points and the resulting dimensions of matrix $\mathbf{A}$ for all experiments in *Appendix* Table 4 (page 23).
>
> Regarding grid dependency, we provide the following clarifications:
>
> **(1) Grid-free Nature and Resolution Adaptability:**
>
> Similar to PINNs and neural operators, our model is grid-free: no mesh construction is required; collocation points can be sampled arbitrarily, and the trained model can give predictions at arbitrary locations, even on finer grids. For example, in the five benchmark cases (*Appendix* Table 4,5; page 23,24), trained on a coarse $32 \times 32$ grid, the model achieves accurate zero-shot generalization on a $128 \times 128$ grid.
>
> We note that the physics-informed Tikhonov-regularization still depends on collocation density—overly sparse sampling weakens PDE enforcement, as in standard PINNs. However, prior studies still suggest that PINNs retain the potential to overcome the curse of dimensionality in solving high-dimensional and fractional PDEs [1,2].
>
> **(2) Theoretical Basis for Grid Independence:**
>
> For a single task (one PDE), the rows of $\mathbf{A}$ and $\mathbf{b}$ equal the total number of collocation points (PDE, IC, and BC), while their columns correspond to the product of the pre-final feature dimension $N_n(L-1)$ and the output dimension $D_o$.
>
> Since the Tikhonov update
> $(\lambda_{\mathrm{reg}}\mathbf{I} +\mathbf{A}^T\mathbf{A})\boldsymbol{w}=\mathbf{A}^T\mathbf{b}$
> involves only $\mathbf{A}^T\mathbf{A}\in\mathbb{R}^{ D_o N_n(L-1)\times  D_o N_n(L-1)}$, the computational cost is governed by the feature dimension rather than the (potentially large) number of collocation points. This keeps the Tikhonov solver tractable as long as the feature dimension is properly controlled. In our experiments, the Newton-PINet achieves high accuracy without requiring a large pre-final feature dimension (i.e., no. features $\ll$ no. collocation points).
>
> [1] Hu et al. Score-based physics-informed neural networks for high-dimensional Fokker–Planck equations. *SIAM J. Sci. Comput.*, 2025.
>
> [2] Pang et al. fPINNs: Fractional physics-informed neural networks. *SIAM J. Sci. Comput.*, 2019.

---

> > ### Author Rebuttal · Reviewer_UYkA · 2026-04-01
> >
> > The reviewer thanks the authors for clarifications. My questions are resolved.

---

> > > ### Author Response · Authors · 2026-04-01
> > >
> > > Thank you for your valuable suggestions and for raising the score. We will improve the final manuscript accordingly. Wishing you all the best in your work.

---

### Official Review · Reviewer_g64x · 2026-03-11

**Soundness:** 4
**Presentation:** 4
**Significance:** 4
**Originality:** 4
**Overall Recommendation:** 6
**Confidence:** 4

**Summary:**

The authors propose a PINN framework for non-linear PDEs, reformulating it as a least squares problem, following Newton linearization scheme. They propose gradient free tuning of the output layer via least squares. For fine tuning on unseen tasks, they employ Tikonov regularization. The authors conduct extensive experiments on a wide range of settings, with a thorough and comprehensive appendix.

**Compliance With Llm Reviewing Policy:**

Affirmed.

**Key Questions For Authors:**

1. Is it possible to re-compute Table 3 on identical environments to ensure fairness in comparisons? It is particularly crucial for training times, although the accuracy and loss numbers should not vary too much.

**Limitations:**

The authors do not provide a discussion on limitations. Some questions remain regarding the fairness in baseline comparisons which should be clarified.

**Strengths And Weaknesses:**

*Strengths*
**Newton Linearization**: The Newton linearization and Tikhonov regularization are very interesting directions to pursue, particularly given the recent focus on improving PINN learning frameworks using linearized PDE forms [1]

**Empirical results and presentation**: The empirical studies are very thorough, rigorous and the presentation is very clean and easy to understand, with the speedup numbers being quite impressive.

**Generalization**: The experiments listed in Appendix D and E, in particular provide valuable insights into the scalability and generalizability of the proposed approach.

*Weaknesses*
**Lack of discussions on limitations** While the authors present a compelling paper, they omit the discussion of limitations altogether. It would be valuable to include a section that discusses the failure modes of the proposed approach.

**Computational Complexity Analysis**: The current work does not include formal analysis of computational complexity of the proposed approach. While this is not necessary, it would help to provide an intuition or a discussion on how the complexity scales in systems with multiple output variables, and multi-physics problems

**Table 3 Clarification**: The results presented in table 3 are not computed in identical environments and therefore seem a bit unfair, especially the training times. I might need more clarifications on this.

Questions:


[1] Operator learning using weak supervision from walk on spheres, Viswanath et al,

---

> ### Author Rebuttal · Authors · 2026-03-31
>
> We sincerely thank the reviewer for recognizing the value of our work.
>
> **Response to W1 (Lack of Discussions on Limitations)**
>
> We have discussed the limitations and future directions of the model in *Appendix E* (page 28–29), but did not elaborate in the main text due to space constraints. We will further expand this discussion and add a brief section in the revised main text.
>
> Briefly, the task adaptation is formulated as a physics-informed Tikhonov-regularization problem, whose tractability mainly depends on the size and conditioning of $\mathbf{A}$ (see *Appendix E* for details). The potential future directions can be summarized as follows:
>
> (1) The matrix $\mathbf{A}$ is dense rather than sparse, since each row encodes features from a multilayer neural representation with skip connections. Its rank and conditioning depend on collocation points, the Tikhonov parameter, and the hidden-layer feature expressiveness. Standard sparse PDE solvers offer limited advantage; designing scalable solvers for dense neural-feature matrices is a promising future direction (*Appendix E*).
>
> (2) Another limitation is GPU memory. On a single 16 GB V100, each task adaptation requires solving the full least-squares matrix, forcing small batch sizes and limiting meta-learning capacity. Future work will use larger, higher-performance GPUs to handle more complex tasks efficiently.
>
> **Failure modes:** Current experiments focus on 1D+time and 2D N-S cases. For 2D/3D+time or multi-physics systems, larger and more ill-conditioned $\mathbf{A}$ matrices combined with GPU limits prevent effective solution. We also note that the method still depends on collocation density—overly sparse sampling weakens PDE enforcement, as in standard PINNs. Additionally, for long-time prediction of chaotic systems such as K-S, we currently require 15 temporal blocks for accurate inference. Enabling single-shot prediction or reducing dependence on temporal decomposition remains an open challenge. **We will clarify these points in the revised manuscript.**
>
> **Response to W2 (Computational Complexity Analysis)**
>
> We have detailed this discussion in *Appendix E* and will better highlight it and add a reference in the main text so that it's clear to the reader. **Please refer to the response to Q1 of Reviewer 1 [Mg3J], where we report an empirical study on inference time for varying numbers of columns in $\mathbf{A}$.**
>
> Briefly, the scalability of the Tikhonov solution depends on the size of matrix $\mathbf{A}$, whose rows scale with the total number of PDE/IC/BC residuals and whose columns scale with the pre-final feature dimension and output dimension $D_o$ (due to skip connections, the feature dimension equals the total neurons in the $L-1$ hidden layers with each of width $N_n$). For 2D/3D or multi-physics systems, more equations and higher output dimensions enlarge $\mathbf{A}$. Since the Tikhonov update
> $(\lambda_{\mathrm{reg}}\mathbf{I} +\mathbf{A}^T\mathbf{A})\boldsymbol{w}=\mathbf{A}^T\mathbf{b}$
> involves only $\mathbf{A}^T\mathbf{A}\in\mathbb{R}^{ D_o N_n(L-1)\times  D_o N_n(L-1)}$, its cost mainly depends on the feature dimension rather than the number of residual samples. This keeps the Tikhonov solver tractable as long as the feature dimension is properly controlled. Nonetheless, as discussed above, matrix conditioning remains challenging for complex 2D/3D multiphysics problems, motivating future improvements in solution strategies.
>
> **Response to W3 and Q1 (Table 3)**
>
> For Table 3 in the manuscript, we further trained and evaluated our model on an A6000-GPU. As shown in the table below, under the same GPU, Newton-PINet still achieves higher test accuracy than the state-of-the-art model (PI-neural-solver [2]), while maintaining significantly better computational efficiency.
>
> A comparison of Newton-PINet on A6000 and V100 GPUs shows that training time differs across the two GPUs, while inference time remains comparable. This is likely because GPU configurations mainly affect the efficiency of gradient-based training, whereas our gradient-free Tikhonov-based adaptation is much less sensitive to such differences.
>
> |                           | Model                        | No. training / test tasks | Test relative MSE | Training time | Inference time |
> |---------------------------|------------------------------|---------------------------|-----------------|---------------|----------------|
> |                           | PI-neural-solver (A6000-GPU) [2] | 800 / 200                | 2.91e-4          | 4h30m         | 0.284s         |
> |                           | **Newton-PINet (A6000-GPU)** | 50 / 200                 | 1.72e-7          | 301s          | 0.080s         |
> |                           | **Newton-PINet (V100-GPU)**  | 50 / 200                 | 1.71e-7          | 119s          | 0.084s         |
>
> [2] Boudec et al. Learning a neural solver for parametric PDEs to enhance physics-informed methods. *ICLR*, 2025.

---

> > ### Author Rebuttal · Reviewer_g64x · 2026-04-01
> >
> > Thank you for the response. I maintain my score.

---

> > > ### Author Response · Authors · 2026-04-01
> > >
> > > We sincerely appreciate your recognition of our work. We will incorporate the relevant updates in the revised manuscript. Wishing you all the best.

---

### Official Review · Reviewer_ajuT · 2026-03-11

**Soundness:** 2
**Presentation:** 3
**Significance:** 3
**Originality:** 2
**Overall Recommendation:** 3
**Confidence:** 3

**Summary:**

The paper proposes a meta-learning framework for solving partial differential equations (PDEs), grounded largely in the physics-informed neural networks (PINNs) formalism. The main contributions include the introduction of skip connections to enhance expressivity, the development of a two-stage training strategy in which the output layer is solved via a least-squares formulation, and the incorporation of Newton linearization to efficiently handle nonlinear PDEs and accelerate convergence during task-specific adaptation.

**Compliance With Llm Reviewing Policy:**

Affirmed.

**Final Justification:**

Thank you to the authors for the detailed and thoughtful rebuttal. I appreciate the effort to acknowledge the concerns and clarify the methodological and experimental choices.

Regarding W2, the additional clarification on the computational cost is helpful, and the planned revision to better reflect the regime-dependent behavior is appreciated.

Regarding W3, I appreciate the authors’ explanation and the challenges associated with reliably re-implementing baselines. However, the current experimental setup still deviates from common practice in top-tier ML venues, where efforts are typically made to include relevant baselines for newly introduced problem settings, or to more clearly contextualize the absence of such comparisons. While I understand the reasoning, this remains a limitation in fairly assessing the proposed method on the new benchmarks.

Overall, while the rebuttal improves clarity and addresses several concerns, I still find the work to fall on the negative side due to the above issue with experimental validation relative to standard practice.

**Key Questions For Authors:**

For the concerns/major questions, please refer to Weaknesses above.

Some additional clarifying questions are:

- Missing rationale on modeling choices: why the concatenation of all hidden representations is expected to be effective?

- Misnomer: are there really skip-connections (which does have x + f(x) type operation)? While there are some ‘highway-type’ connections, there are no connections that can be considered as ‘skip-connections’.

**Limitations:**

The authors does not have an explicit section/paragraph to discuss limitations. As elaborated in Weaknesses, the scalability for high-dimensional PDEs could be the main concern/limitations.

There is no potential negative societal impact expected.

**Strengths And Weaknesses:**

Strengths:
- Neural PDE solvers / neural surrogate models for PDEs are important areas and the topic of this paper is overall well-aligned with those of ICML.

- The proposed method seems to achieve improved performance for the benchmark problems in the considered scenarios.

- The paper includes some necessary ablation studies to investigate several aspects of the proposed method (e.g., the number of Newton iterations, network depth/width/activation/weight initialization/learning rate).

Weaknesses:
- Novelty regarding main contributions: (1) skip-connections: while the specific setup of the skip-connections studied in this paper has a very particular form (i.e., connections from all hidden layers to the output layer), the idea of skip-connections has been studied in the literature (See [1][2]). (2) least-squares solve for the output layer: while the current method introduce additional contribution (i.e., Newton linearization), overall the proposed training algorithm resembles with the Least-squares Gradient Descent (LSGD) algorithm shown in [3] --- indeed, the LSGD paper presented sophisticated way to combining LS and GD along with detailed analysis and specialized initialization schemes. The authors need to clearly address which part of the contributions are really new considering those previous works. Relatedly, the idea of adopting only the last layer is a lot discussed in the meta-learning context, which can be exemplified by a method called, ANIL (Almost No Inner Loop), in [4]. Some discussion on this would be needed.

- The scalability could be a major concern: as the authors acknowledge in their discussion section, increased dimensionality would require more collocations points and usually require larger-sized neural networks (larger width/depth); the increase in size of A could potentially lead to a non-practical method.

- Some non-standardized ways of comparing against baselines: the paper seems to utilize the results reported from the previously reported in the references and, due to this reason, there are many empty entries in tables (e.g., Table 2). Is this correct understanding? With this setup, only a partial picture seems to be shown. Relatedly, some benchmark PDEs are compared with the number of baselines while other PDE benchmarks are only considered with DNNs and PINet. Moreover, it is confusing if comparing Hybrid methods against Unsupervised methods provides any meaningful values. For example, in Table 3 (reaction-diffusion results), it is expected that there will be a big difference between the methods that are using data and not using data.

[1] Kim, et al, DPM: A novel training method for physics-informed neural networks in extrapolation, AAAI, 2021

[2] Wang, et al, PIRATENETS: PHYSICS-INFORMED DEEP LEARNING WITH RESIDUAL ADAPTIVE NETWORKS, JMLR, 2024

[3] Cyr, et al, Robust Training and Initialization of Deep Neural Networks: An Adaptive Basis Viewpoint, MSML, 2020.

[4] Raghu, et al, Rapid Learning or Feature Reuse? Towards Understanding the Effectiveness of MAML, ICLR, 2020.

---

> ### Author Rebuttal · Authors · 2026-03-31
>
> Thank you for your constructive comments. We clarify our novelty, scalability, and benchmarking study below.
>
> **Response to W1:**
>
> (1) Our design is different from standard residual connections (i.e., $x+f(x)$) used in prior works [1,2] for improving gradient flow. Instead, we concatenate features from all hidden layers and feed them into the output layer, followed by a least-squares solve. Our main goal is to construct a richer basis space, where each hidden layer progressively contributes to additional nonlinear features. This enhances representation capacity and improves the conditioning and accuracy of the physics-informed Tikhonov regularization solution, rather than merely stabilizing training.
>
> (2) While LSGD [3] also combines least-squares and gradient descent, it focuses on data-driven single-task learning with linear structures. Our novelty lies in extending this framework to meta-learning for nonlinear PDEs via physics-informed Tikhonov-regularized least-squares. Unlike [3], our system is explicitly built from physical constraints (PDE residuals, ICs, and BCs), and Newton linearization converts nonlinear residuals into tractable local linear least-squares problems. This enables efficient, gradient-free inner-loop adaptation that inherently enforces physical laws—entirely absent in [3].
>
> (3) Our method shares the idea of updating only the last layer with ANIL [4] but differs in the optimization paradigm. Rather than using gradient-based updates, we perform closed-form, physics-constrained least-squares solves with Tikhonov regularization. This eliminates inner-loop gradient descent and exploits the algebraic structure of PDEs, enabling fast physics-informed learning on new PDE configurations—fundamentally distinct from conventional data-driven meta-learning.
>
> **In summary:** Solving complex nonlinear PDEs via meta-learning presents fundamental challenges that require entirely different algorithmic considerations from traditional data-driven methods like LSGD or ANIL. By synergistically integrating basis-enriching skip-connections, Newton linearization, and physics-constrained Tikhonov least-squares, our work proposes a completely new learning paradigm tailored to the algebraic structure of physical systems, which is firmly backed by both theoretical and empirical analyses. We appreciate the reviewer’s suggestion to highlight the distinctions between our work and prior studies, and **we will incorporate above discussion into the revised Related Work section.**
>
> **Response to W2 and Limitations:**
>
> We provide limitation and mitigation discussion in *Appendix E* (page 28-29). Also kindly refer you to our responses to W1 and W2 of **Reviewer 3 [g64x]**, which address a similar concern.
>
> Briefly, the cost of Tikhonov update mainly depends on the feature dimension rather than the number of residual samples. This keeps the Tikhonov solver tractable as long as the feature dimension is properly controlled.
>
> **Response to W3:**
>
> (1) Your understanding is correct. We cite originally reported metrics and leave unreported ones blank to ensure fair comparisons and prevent reimplementation bias.
>
> (2) Standard PDEs have many baselines. However, for CDR, KdV, and K-S, we introduced challenging new datasets with varying initial conditions. Lacking prior evaluations on these complex settings, only DNN and PINet serve as direct baselines.
>
> (3) We agree hybrid methods naturally win. Our intention in Table 3 is not to claim superiority, but to highlight that purely physics-based unsupervised training can struggle with highly nonlinear problems. This motivates the use of a small amount of labeled data to improve the representation learning prior to test-time inference. **We will make this explicit in the revised manuscript.**
>
> **Response to Questions:**
>
> The rationale is detailed in the second paragraph of *Appendix A.1* (page 11). In brief, our design constructs a richer polynomial/Chebyshev-type basis, improving expressivity and the stability of the Tikhonov solution. Through extensive ablation studies (*Appendix D.6*; page 27), we demonstrate that model benefits from this architecture, which enhances the model's robustness to network depth, width, and mesh resolution, thereby reducing the need for extensive hyperparameter tuning.
>
> Our “skip-connections” denote concatenating features from all hidden layers to the output layer (not $x+f(x)$), enhancing the Tikhonov representation. This terminology aligns with the broader deep learning consensus, where topological bypasses via feature concatenation are formally established as skip-connections in foundational architectures like DenseNet [5] and Hypercolumns [6].
>
> [5] Huang et al. Densely connected convolutional networks. *CVPR*, 2017.
>
> [6] Hariharan et al. Hypercolumns for object segmentation and fine-grained localization. *CVPR*, 2015.

---

> > ### Author Rebuttal · Reviewer_ajuT · 2026-04-04
> >
> > Thank you the authors for providing the rebuttal.
> >
> > **W1**
> >
> > Thank you for agreeing to include the discussion regarding W1.
> >
> > **W2**
> >
> > First, I'd like to clarify that the "discussion section" mentioned in the original comment refers to Appendix E. The reported inference time (in response to Q1 for Mg3J) provides some useful intuition on scaling, which is very helpful. However, the authors may need to be more careful with the statement that ``the cost of Tikhonov update mainly depends on the feature dimension rather than the number of residual samples.'' There are important regimes, for example high dimensional or parametric PDEs, where the number of residual samples $m$ must be very large. In such cases, the cost of forming the matrix $A$ and computing $A^\top A$ can dominate, and may even become a bottleneck relative to solving the resulting system. It would be beneficial to say explicitly that this method scales well on such problems. I'd appreciate the authors' response on this.
> >
> > **W3**
> >
> > The lack of baselines on the newly introduced problems is a concern. While there may not be an explicit rule, it is generally common practice in top-tier ML venues like ICML that when new problem settings or datasets are introduced, authors attempt to implement and evaluate relevant existing methods as baselines, or clearly document limitations if this is not feasible. At the same time, I acknowledge that there may not be a universally agreed-upon standard here, and reasonable arguments could be made on both sides. I therefore do not intend to overemphasize this point, and leave it to the authors to clarify their choices and to the AC to weigh its importance.

---

> > > ### Author Response · Authors · 2026-04-07
> > >
> > > **Response to W1:**
> > >
> > > We appreciate the reviewer's acknowledgement of our response to W1.
> > >
> > > **Response to W2:**
> > >
> > > We thank the reviewer for this careful observation. We agree that the statement regarding the cost of the Tikhonov update should be made more carefully.
> > >
> > > In our formulation, the column dimension of $\mathbf{A}$ scales with the no. (number of) features and equations (i.e., output variables), while the row dimension scales with no. residual points $m$ and equations.
> > >
> > > The cost of the linear solve (after forming $\mathbf{A}$) depends on the smaller dimension, i.e., $\min(\text{rows}, \text{columns})$. In contrast, the cost of constructing $\mathbf{A}$ and $\mathbf{A}^T \mathbf{A}$  depends on the larger dimension, i.e., $\max(\text{rows}, \text{columns})$, and can dominate when $m$ is very large. Therefore, the overall computational cost is regime-dependent. However, in our framework, $m$ need not be large even for high-dimensional or parametric PDEs, and tasks can be efficiently mini-batched.
> > >
> > > (1) For certain high-dimensional systems (e.g., multi-variable ODEs with time as the only input), our model remains efficient even with many equations (e.g., $\sim$100), as long as a moderate number of collocation points is needed. When the number of collocation points increases, scalability can still be maintained by controlling the feature dimension. In practice, these factors can be flexibly balanced under memory constraints.
> > >
> > > (2) For parametric PDEs, the parameters define different tasks and can be large but they are not directly used as network inputs. Instead, our model learns shared representations across tasks. During training, we avoid sampling residual points across parameter inputs (as in P$^2$INNs [1]), and instead construct $\mathbf{A}$ directly from spatiotemporal points for least-squares adaptation. So, the number of residual samples depends only on the spatiotemporal dimension, not the parameter dimension, which avoids unnecessary growth in the size of $\mathbf{A}$.
> > >
> > > As the reviewer pointed out, in high-dimensional regimes, the cost of forming $\mathbf{A}$ and $\mathbf{A}^T \mathbf{A}$ may become dominant. Nevertheless, even in such cases, our gradient-free adaptation can be more efficient than gradient-based fine-tuning in PINNs.
> > >
> > > **We will revise the manuscript to clarify the computational cost of the Tikhonov update and discuss the applicability and limitations of our method in high-dimensional and parametric PDE settings.** We appreciate this valuable suggestion.
> > >
> > > [1] Cho et al. Parameterized physics-informed neural networks for parameterized PDEs. ICML, 2024.
> > >
> > >
> > > **Response to W3:**
> > >
> > > Thank you for this fair point. We agree that adapting related methods as baselines is valuable and generally expected when feasible. Our benchmarking strategy is to evaluate our method on well-established benchmark problems and compare against previously reported results, which we believe enables a fairer comparison while avoiding under-representing prior methods. Accordingly, Tables 2,3 (page 7) compare with several state-of-the-art neural operators, including PINO (2024) [2], T-DeepONet-TF (2025) [3], and PI-Neural-Solver (2025) [4]. Table 6 (Appendix, page 26) further reports results on four nonlinear PDEs against eight recent meta-learning PINN methods. These results consistently show the advantage of our model, with better generalization (requiring orders of magnitude fewer training tasks) and substantially lower training cost (greater task efficiency and faster inference) than existing neural-operator methods.
> > >
> > > For the newly introduced, more complex parametric PDE benchmarks (e.g., varying IC/coefficient CDR, KdV, and chaotic K-S), we only include comparisons among our model variants. We did not include additional baselines from the literature, as our re-implementations were not reliable without extensive tuning. To avoid bias from suboptimal reproductions, we chose not to report these results. Nevertheless, we believe that evaluations on widely studied benchmarks already provide strong evidence of the model’s effectiveness. **We will explicitly clarify this limitation in the revised manuscript and plan to release these challenging benchmarks to support future research.**
> > >
> > > We also wish to highlight that our experimental study is broad and extensive, including comparisons with related work, multiple newly introduced problems, detailed ablations, and thorough analysis of key factors such as the no. Newton iterations, no. residual samples, network depth/width, activation functions, weight initialization, and learning rate.
> > >
> > >
> > > [2] Li et al. Physics-informed neural operator for learning partial differential equations. ACM/IMS Journal of Data Science, 2024.
> > >
> > > [3] Wei et al. Efficient transformer-inspired variants of physics-informed deep operator networks. arXiv preprint, 2025.
> > >
> > > [4] Boudec et al. Learning a neural solver for parametric PDEs to enhance physics-informed methods. ICLR, 2025.

---

### Official Review · Reviewer_Mg3J · 2026-03-12

**Soundness:** 3
**Presentation:** 3
**Significance:** 3
**Originality:** 3
**Overall Recommendation:** 5
**Confidence:** 2

**Summary:**

The paper proposes Newton-PINet, a novel meta-learning framework designed to efficiently solve parameterized nonlinear PDEs. The architecture utilizes a multi-layer network with skip connections, where the hidden layers are optimized via gradient descent during a meta-learning phase to learn robust representations across a family of PDEs. For rapid task-specific adaptation, the output layer weights are computed using gradient-free Tikhonov regularization least squares. The core methodological contribution is the replacement of the linearly convergent Picard iteration with Newton linearization within this least-squares framework. This integration allows the model to achieve quadratic convergence during the gradient-free fine-tuning stage for nonlinear terms. Extensive empirical evaluations across various 1D and 2D PDEs demonstrate that Newton-PINet outperforms SOTA physics-informed and data-driven baselines in both data efficiency and computational speed.

**Compliance With Llm Reviewing Policy:**

Affirmed.

**Key Questions For Authors:**

1. Scalability of Dense Inversions: In Appendix E, you acknowledge that scaling to high-dimensional outputs ($D_o$) or multi-physics systems increases the columns of matrix A, making the dense matrix inversion $A^T A$ more computationally demanding. Could you provide an empirical sense (e.g., a scaling plot or complexity analysis) of how the wall-clock inference time grows as $D_o$ increases?
2. Temporal Decomposition Limits: For chaotic systems like K-S, your method required 15 time blocks to maintain a valid initial guess for the Newton iterations. Does the error compound significantly over these blocks for longer time horizons? Have you observed a theoretical or empirical limit to how far this temporal domain decomposition can be pushed before the initial guess $u_{guess}$ completely diverges?
3. Objective Conflict in Meta-Learning: You noted that combining physics-based and data-driven meta-learning losses ($l_{LSE+MSE}$) sometimes generalizes worse than pure $l_{MSE}$ due to objective conflict. In a pure $l_{MSE}$ meta-training setup, do we risk losing the "physics-informed" regularizing effect in the latent representation space, effectively reducing the outer loop to a standard data-driven operator?

**Limitations:**

Yes.

**Strengths And Weaknesses:**

Strengths: The mathematical foundation of the paper is solid. The paper is highly readable and logically structured. The paper addresses a well-known and highly frustrating bottleneck in the AI for Science community: the prohibitive training time and complex loss landscapes associated with PINNs for nonlinear PDEs.
Weaknesses: The proposed gradient-free fine-tuning heavily relies on a high-quality initial guess for the nonlinear iterations, particularly for chaotic systems like the K-S equation. To achieve convergence on such systems, the authors had to rely on a heavy 15-block temporal domain decomposition. This somewhat restricts the "single-shot" elegance of the solver for chaotic dynamics over long time horizons.

---

> ### Author Rebuttal · Authors · 2026-03-31
>
> **Response to Weaknesses:**
>
> We sincerely thank the reviewer for recognizing the value of our work. For complex time-dependent PDEs, training PINNs over the entire spatialtemporal domain in a single-shot manner usually fails. As noted in prior work [1], gradient-based training tends to fit later-time dynamics before resolving initial conditions, which can lead to premature convergence (and incorrect solutions), especially for chaotic systems such as K-S. Consequently, many recent PINNs adopt temporal decomposition, typically at the expenses of high computational cost.
>
> Separately, many neural PDE solvers (e.g., neural operators) [2–4] also rely on autoregressive (one-step) inference for long-time prediction. This strategy is consistent with classical numerical methods, where solutions are advanced sequentially in time to ensure stability.
>
> Therefore, for complex time-dependent PDEs, temporal decomposition is a practical and effective strategy across both AI-based and classical methods. Nonetheless, enabling stable single-shot PINNs remains an important direction, and we appreciate the reviewer’s insightful comment on their potential.
>
> [1] Wang et al. An expert's guide to training physics-informed neural networks. *arXiv*, 2023.
>
> [2] Hou et al. Learning neural operators from partial observations via latent autoregressive modeling. *arXiv*, 2026.
>
> [3] Li et al. Learning dissipative dynamics in chaotic systems. *arXiv*, 2021.
>
> [4] Han et al. Predicting physics in mesh-reduced space with temporal attention. *arXiv*, 2022.
>
> **Response to Q1:**
>
> We conduct experiments on the 2D steady Navier-Stokes equations for the lid-driven cavity (LDC) problem to evaluate how inference time varies with $D_o$. The number of columns of $\mathbf{A}$ is $D_o N_n (L-1)$, equal to the product of the pre-final feature dimension (i.e., the total number of neurons in the $L-1$ hidden layers with each of width $N_n$ due to skip connections) and the output dimension $D_o$. For this LDC case, $N_n = 128$ and $L = 5$ are fixed, $D_o = 3$ corresponds to predicting $(u, v, p)$, while $D_o = 2$ and $D_o = 1$ correspond to predicting $(u, v)$ with known pressure $p$, and $u$ with known $(p, v)$, respectively. Table 1 reports the inference time for different $D_o$.
>
> Since the number of columns of $\mathbf{A}$ is $D_o N_n (L-1)$, we further examine the cost increase due to the column size by varying $N_n$ (Table 2). The results show that increasing either $D_o$ or $N_n$ enlarges $\mathbf{A}$, thus increasing the cost of the Tikhonov regularization solve and task adaptation time.
>
> *Table 1: Inference time versus $D_o$, with $N_n = 128$ and $L = 5$ fixed.*
> |                              | $D_o=1$ | $2$  | $3$  |
> |------------------------------|--------|------|------|
> | Number of **A** columns      | 512    | 1024 | 1536 |
> | Inference time (s)           | 0.25   | 0.31 | 0.40 |
>
>
> *Table 2: Inference time versus $N_n$, with $D_o = 3$ and $L = 5$ fixed.*
> |                              | $N_n=64$ | $128$  | $256$  | $512$  |
> |------------------------------|----------|------|------|------|
> | Number of **A** columns      | 768      | 1536 | 3072 | 6144 |
> | Inference time (s)           | 0.29     | 0.40 | 0.76 | 2.11 |
>
> **Response to Q2:**
>
> Regarding the mitigation of error propagation over time, we summarize several effective strategies:
>
> (1) Iteration and decomposition control: Increasing the number of nonlinear iterations within each time block, or using more time blocks, improves accuracy but increases computational cost (see Table 9 in Appendix D.3).
>
> (2) Improved initialization: Using the exact initial condition (IC) as the initial guess for the first block, followed by sequential inference, helps reduce error amplification.
>
> (3) LSE-based thresholding: Since task-adaptation is based on Tikhonov regularization, the least-squares error (LSE) in each block can be used to monitor error growth, e.g., proceeding to the next block only when the LSE falls below a threshold.
>
> (4) IC weighting: Increasing the IC penalty weight in the Tikhonov regularization ($\lambda_{\mathrm{ic}}$) can effectively reduce boundary drift errors between time blocks (see Table 10 in Appendix D.4).
>
> These strategies provide practical ways to mitigate temporal error accumulation, which we plan to further explore to improve robustness and performance.
>
> **Response to Q3:**
>
> As illustrated in Fig. 1 and the pseudocode in the Appendix (page 14–15), both the inner-loop adaptation in meta-learning and the inference stage minimize the physics loss (LSE) via gradient-free Tikhonov regularization to update the final-layer weights. The outer loop in meta-learning updates the hidden layers via gradient descent on the data loss (MSE), even as predictions are always obtained through the Tikhonov regularization that enforces PDE constraints. Consequently, our meta-learning stage is not purely data-driven but inherently physics-informed due to the physics-based inner-loop adaptation.

---

> > ### Author Rebuttal · Reviewer_Mg3J · 2026-04-01
> >
> > Thank you for the clarification. The additional explanations resolve my main concerns. Since my initial rating was already positive, I will maintain my score and look forward to seeing these updates in the final version.

---

> > > ### Author Response · Authors · 2026-04-01
> > >
> > > Thank you for your acknowledgment and recognition of our work. We will update the corresponding content in the revised manuscript. Wishing you all the best in your work !

---

### Decision · Program_Chairs · 2026-04-30

**Decision:**

Reject

**Comment:**

While reviewers agreed that the paper addresses common problems in PINNs with prohibitive training and complex loss landscapes, they also raised serious concerns about the paper's novelty and empirical soundness.  While the Newton linearization addresses the scalability problem in nonlinear equations, a similar approach was also done in ProbHardE2E (ICLR 2026). There are also concerns on the clarity of presentation of the paper. A major concern on the evaluation is the missing baselines in Table 2 and I agree that it is common practice to reimplement these baselines, e.g., FNO to ensure a reproducible and fair comparison as is common practice at top-tier ML conference venues. The table is also difficult to interpret and draw conclusions from given the methods are evaluated on different test set sizes. Therefore, the paper in its current state cannot be accepted to the conference at this time. We encourage the authors to revise, include the additional missing baselines and resubmit to an appropriate future venue.